# BLOCKWISE SFT FOR DIFFUSION LANGUAGE MODELS: RECONCILING BIDIRECTIONAL ATTENTION AND AUTOREGRESSIVE DECODING

## ABSTRACT

Discrete diffusion language models have shown strong potential for text generation, yet standard supervised fine-tuning (SFT) misaligns with their semi-autoregressive inference: training randomly masks tokens across the entire response, while inference generates fixed-size blocks sequentially. This mismatch introduces noisy prefixes and leaky suffixes, biasing gradients away from the desired blockwise likelihood. We propose Blockwise SFT, which partitions responses into fixed-size blocks, selects one active block per step for stochastic masking, freezes all preceding tokens, and fully hides future ones. Loss is computed only over the active block, directly mirroring the blockwise decoding process. Experiments on GSM8K, MATH, and MetaMathQA show consistent gains over classical SFT under equal compute or token budgets. Block size consistency studies and ablations confirm that improvements stem from faithful training–inference alignment rather than incidental masking effects. Our results highlight the importance of matching supervision granularity to the decoding procedure in diffusion-based language models.

## 1 INTRODUCTION

Discrete diffusion models have emerged as a promising alternative to autoregressive language models for text generation. Unlike traditional left-to-right decoding, diffusion language models (LMs) learn to iteratively denoise corrupted text by predicting clean tokens from noisy versions (Gong et al., 2023). During training, these models corrupt response tokens with a relaxed categorical process and learn to recover the original text; during inference, they start from pure noise and progressively refine it into coherent text through multiple denoising steps.

Recent advances have explored semi-autoregressive decoding as a practical compromise between fully parallel generation and strict sequential processing. Systems like SSD-LM generate text in fixed-size blocks, where tokens within each block are produced in parallel while maintaining causal dependencies across blocks (Han et al., 2023). Modern discrete diffusion LMs, including the LLaDA family, have adopted blockwise inference as their primary generation strategy (Nie et al., 2025). This widespread adoption of blockwise decoding raises a fundamental question: *can a training objective based on bidirectional attention and full-sequence random masking effectively serve a model that will be decoded causally, block by block?*

The prevailing supervised fine-tuning (SFT) approach for diffusion LMs is fundamentally misaligned with blockwise inference. Classical SFT randomly masks tokens across the entire response and reconstructs them using bidirectional context in a single pass. However, during inference, the model operates differently: it receives a clean, deterministic prefix and must produce exactly one block while all future tokens remain completely hidden. As illustrated in figure 1, this training-inference mismatch creates three critical problems: (i) noisy prefixes: the model trains on corrupted contexts that never appear during generation; (ii) dependency leakage: training may reveal tokens within or beyond the target block, violating the causal constraints enforced at inference; and (iii) granularity mismatch: training optimizes token-level decisions while inference requires block-level coordination.

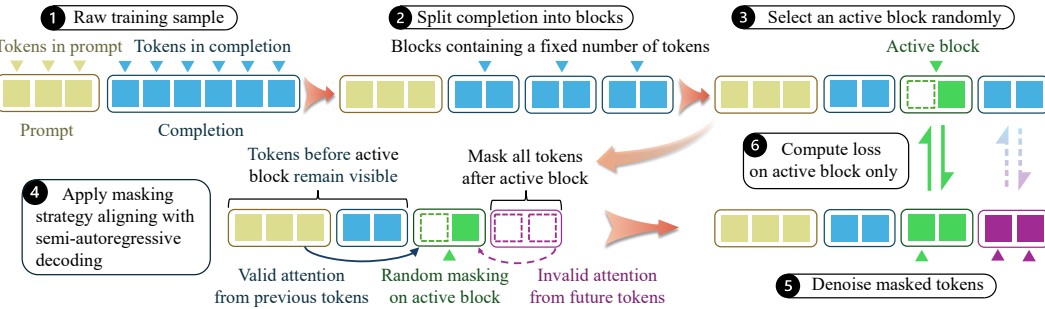

Figure 1: Mismatch between training and inference: Classical SFT uses bidirectional attention and full-sequence masking, exposing future tokens and corrupting parts of the prefix; semi-autoregressive decoding is one-directional, conditioning only on a clean prefix while future blocks are strictly hidden.

To address this misalignment, we introduce Blockwise SFT (Figure 2), a training objective that mirrors deployment: at each step we supervise a single active block, keep the prefix clean and frozen, hide all future tokens, and compute loss only within the active block. This aligns gradients with blockwise decoding, preserves cross-block causality, and concentrates supervision where decisions are actually made. The recipe is architecture-agnostic and applies to any diffusion LM that supports blockwise inference.

Figure 2: Blockwise SFT step: split the completion, sample an active block, keep a clean prefix and a hidden future, and train only on masked tokens inside that block.

**Contributions.** (1) We diagnose the core mismatch between classical full-response SFT and blockwise semi-autoregressive decoding, showing how global random masking induces noisy prefixes and suffix leakage. (2) We propose Blockwise SFT, a drop-in objective that supervises one active block at a time under a clean prefix and hidden future, requiring no architectural or sampling changes. (3) We provide theory: a variational upper bound on blockwise likelihoods, unbiased timestep-sampled gradients, and a characterization of the gradient bias in classical SFT under mismatch.

Aligning training with inference yields concrete gains. Fine-tuning on MetaMathQA and evaluating on GSM8K and MATH, Blockwise SFT consistently improves Pass@1 under matched FLOPs and matched supervised tokens, with faster early-training progress. Performance peaks when the training block size matches the inference block size and drops with injected prefix noise or suffix visibility, corroborating our alignment analysis.

## 2 RELATED WORK

**Discrete Diffusion Models for Text Generation** Diffusion has been extended to discrete text via Argmax Flows and Multinomial Diffusion, which map tokens to continuous space and add categorical noise during training (Hoogeboom et al., 2021). D3PMs generalize this with structured corruption processes to better capture token relationships (Austin et al., 2023). DiffuSeq applies diffusion to conditional sequence-to-sequence tasks, matching or surpassing autoregressive base-

lines (Gong et al., 2023). Recent evaluations highlight trade-offs between modeling quality and generation speed (Weligalle, 2025).

**Supervised Fine-Tuning Methods** For autoregressive models, SFT trains on cross-entropy loss over response tokens (Ouyang et al., 2022), with variants such as instruction tuning improving generalization (Wei et al., 2023; Chung et al., 2022). Diffusion-based models use iterative refinement, with SFT approaches reconstructing randomly masked tokens in one step (Austin et al., 2023; Li et al., 2022). This departs from the blockwise, semi-autoregressive decoding common in diffusion LLMs, creating a training–inference mismatch.

**Autoregressive Decoding in Diffusion-Based Language Models** Early discrete diffusion generators were slow and sometimes incoherent. SSD-LM introduced semi-autoregressive decoding: generating fixed-size blocks with diffusion denoising while freezing prior blocks (Han et al., 2023). Block Diffusion interpolates between pure diffusion and fully autoregressive sampling via block decomposition and KV-caching, improving flexibility and throughput (Arriola et al., 2025). Adaptive Parallel Decoding (APD) adjusts block size by uncertainty, combining diffusion marginals with an autoregressive verifier to approach AR speeds with minimal quality loss (Israel et al., 2025).

**Contemporary diffusion–SFT variants.** Recent variants emphasize efficiency or stability rather than decoding alignment: MDLM mixes masked-LM–style losses to narrow the gap to autoregressive models (Sahoo et al., 2024); Soft-Masked Diffusion LM uses linguistically informed soft masking with per-step cross-entropy to reduce cost (Chen et al., 2023); RDM reparameterizes the discrete process to improve training and sampling (Zheng et al., 2024); and Two-Step Loss & Scheduling adopts a two-step diffusion with gradually increased self-conditioning to mitigate mismatch (Asada & Miwa, 2025). These methods chiefly modify the objective, schedule, or parameterization while supervising across the full response. In contrast, Blockwise SFT aligns supervision with semi-autoregressive blockwise decoding (clean prefix, hidden future, loss on the active block), yielding consistent gains on GSM8K and MATH under equal compute (Table 1).

## 3 METHODOLOGY

### 3.1 OVERVIEW: FROM TRAINING-INFERENCE MISMATCH TO BLOCKWISE ALIGNMENT

Semi-autoregressive diffusion models face a fundamental challenge: they are trained to denoise tokens anywhere in a sequence, but deployed to generate specific blocks conditioned on clean prefixes. This section develops Blockwise SFT to bridge this gap.

Consider generating a simple expression like "$5 - 3 = ?$". During training, classical methods might mask the "5" or reveal the answer "2" in the suffix, but scenarios that never occur during inference, which always starts from a pristine prefix and generates the next block without future information. This mismatch creates three specific issues we quantify in Section 3.2: (i) training corrupts prefixes that inference keeps clean, (ii) training leaks future tokens that inference never sees, and (iii) training spreads supervision across all positions while inference focuses on the next block.

**Our solution: blockwise alignment.** We make training mirror deployment. At each step, we identify the active block, which is the next chunk the model would generate during inference. We keep its prefix exactly as it appears at deployment (clean and fixed), completely hide the suffix (no future leakage), and concentrate the training signal solely on this active block. This alignment ensures that gradient updates directly optimize the objective used during generation.

**Roadmap.** Section 3.2 analyzes classical SFT's limitations under blockwise decoding and quantifies the resulting gradient bias. Section 3.3 derives our blockwise objective from first principles, proving it provides an unbiased variational bound on the true decoding risk. Section 3.4 translates the theory into a practical algorithm. Throughout, we interleave formal results with intuitive explanations to maintain clarity.

### 3.2 CLASSICAL SFT: ANALYSIS OF TRAINING-INFERENCE MISMATCH

We first recall the classical approach, then characterize how it diverges from blockwise decoding.

**Classical SFT recap.** Given an instruction–response sequence $\mathbf{x}_{1:L} = [\mathbf{c}; \mathbf{r}]$, classical SFT samples independent masks on response tokens with rate $\pi$ (i.e., $m_i \sim \text{Bernoulli}(\pi)$ for $i > L_c$) and trains the model to reconstruct only the masked positions at each diffusion step $t$:

$$\mathcal{L}_{\text{SFT}}(\theta) = \sum_{t=1}^{T} \omega_t \, \mathbb{E}_{\mathbf{x},\mathbf{m},\mathbf{z}_t} \Big[ -\sum_{i>L_c} \mathbf{1}[m_i = 1] \, \log p_\theta\big(x_i \mid \mathbf{z}_t, t\big) \Big]. \tag{1}$$

This objective treats masked positions uniformly and leverages bidirectional context—a capability unavailable during blockwise generation.

**Quantifying the mismatch.** At inference, the model generates block $a$ (indices $\mathcal{I}_a$) conditioned on a clean prefix $\mathcal{I}_{\text{prefix}}^{(a)}$ and without access to the future $\mathcal{I}_{\text{suffix}}^{(a)}$. Classical training violates all three conditions:

- *Prefix corruption.* Random masking almost surely corrupts long prefixes (e.g., with $\pi{=}0.3$ and 100 prefix tokens, corruption occurs with probability $> 1 - 3.3 \times 10^{-16}$), whereas inference always conditions on a pristine prefix.
- *Suffix leakage.* A nontrivial fraction of future tokens remains visible during training even for modest suffix lengths, enabling shortcuts that are disallowed at inference.
- *Diluted supervision.* The loss in Eq. equation 1 spreads gradients over all response positions. The active block obtains only a $|\mathcal{I}_a|/L_r$ share of the total signal, which is identical to any other block despite being the sole target at inference.

These mismatches induce a systematic bias in the learning signal:

**Theorem 3.1** (Gradient bias under training-inference mismatch). *Let $\nabla_\theta \ell^\star(\theta; \mathbf{x}, a)$ denote the ideal gradient for generating block $a$ from a clean prefix (as in inference), and $\nabla_\theta \ell^{cls}(\theta; \mathbf{x}, a)$ the gradient from classical SFT. Suppose the gradient, viewed as a function of the masking pattern, is $L_{pre}$-Lipschitz with respect to the Hamming distance over prefix positions and $L_{suf}$-Lipschitz with respect to the Hamming distance over suffix positions (a bounded-difference condition). Then, taking expectation over the classical SFT masking distribution,*

$$\big\| \, \mathbb{E}\big[\nabla_\theta \ell^{cls}(\theta; \mathbf{x}, a)\big] - \nabla_\theta \ell^\star(\theta; \mathbf{x}, a) \, \big\| \; \leq \; L_{pre} \cdot \pi \, \big|\mathcal{I}_{\text{prefix}}^{(a)}\big| \; + \; L_{suf} \cdot (1-\pi) \, \big|\mathcal{I}_{\text{suffix}}^{(a)}\big|. \tag{2}$$

*In particular, the right-hand side scales linearly with the expected number of corrupted prefix tokens $\mathbb{E}[\#\textit{prefix-corr}] = \pi|\mathcal{I}_{\text{prefix}}^{(a)}|$ and the expected number of visible suffix tokens $\mathbb{E}[\#\textit{suffix-vis}] = (1-\pi)|\mathcal{I}_{\text{suffix}}^{(a)}|$.*

*Intuition.* The bound quantifies how prefix corruption and suffix visibility pull gradients away from the true objective; the bias now scales with the *expected* magnitude of perturbation rather than just the event that at least one token is affected. As sequence length grows, both expected counts increase and the bias cannot be removed by tuning $\pi$, motivating a structurally aligned training procedure. (Proof in Appendix A.6.)

### 3.3 DERIVING THE BLOCKWISE OBJECTIVE

We now develop an objective that aligns training with blockwise inference and admits unbiased gradient estimates for its diffusion-based surrogate.

**Target: blockwise generation risk.** Semi-autoregressive decoding generates a response as $M$ consecutive blocks $\mathbf{b}^{(1)}, \ldots, \mathbf{b}^{(M)}$ following instruction $\mathbf{c}$. The likelihood factorizes as

$$p_\theta(\mathbf{x}) = \prod_{a=1}^{M} p_\theta\big(\mathbf{b}^{(a)} \mid \mathbf{context}^{(a)}, t{=}0\big), \tag{3}$$

where $\mathbf{context}^{(a)} = \mathbf{x}_{1:L_c+B(a-1)}$ represents the clean prefix through block $a{-}1$. The risk we minimize is

$$\mathcal{R}_{\text{block}}(\theta) = \mathbb{E}_{\mathbf{x}} \Big[ -\sum_{a=1}^{M} \log p_\theta\big(\mathbf{b}^{(a)} \mid \mathbf{context}^{(a)}, t{=}0\big) \Big]. \tag{4}$$

This is the exact objective optimized during blockwise inference.

**Constructing a faithful training surrogate.** Direct optimization of Eq. equation 4 is intractable due to the discrete diffusion process. We derive a surrogate that maintains three critical properties: it upper-bounds the true risk, admits efficient stochastic gradient estimation, and preserves the blockwise structure.

For active block $a$ with indices $\mathcal{I}_a$, we apply diffusion training only within this block while keeping the prefix clean and suffix hidden as in $\textbf{context}^{(a)}$. We further sample an intra-block mask $\mathbf{m}$ with entries $m_i \sim \text{Bernoulli}(\pi)$ for $i \in \mathcal{I}_a$, and compute loss only on masked positions:

$$\tilde{\mathcal{L}}_t(\theta; \mathbf{x}, a, \mathbf{m}) = -\sum_{i \in \mathcal{I}_a} \mathbf{1}[m_i = 1] \log p_\theta\big(x_i \mid \mathbf{z}_t, t; \textbf{context}^{(a)}\big), \tag{5}$$

where $\mathbf{z}_t \sim q_t(\cdot \mid \mathbf{x})$ incorporates noise at diffusion step $t$. The full objective averages over blocks, diffusion steps, and masks:

$$\mathcal{L}_{\text{BW-SFT}}(\theta) = \mathbb{E}_{\mathbf{x}} \, \mathbb{E}_{a \sim \rho}\Big[\sum_{t=1}^{T} \omega_t \, \mathbb{E}_{\mathbf{m}, \, \mathbf{z}_t}\big[\tilde{\mathcal{L}}_t(\theta; \mathbf{x}, a, \mathbf{m})\big]\Big], \tag{6}$$

where $\rho$ is a sampling distribution over block indices (uniform in our experiments).

This construction provides rigorous guarantees connecting training to inference via a blockwise variational bound.

**Theorem 3.2** (Variational bound on blockwise generation). *For appropriate weights $\{\omega_t\}_{t=1}^{T}$, the blockwise diffusion objective upper-bounds the generation risk:*

$$-\log p_\theta\big(\mathbf{b}^{(a)} \mid \textbf{context}^{(a)}, t=0\big) \leq \sum_{t=1}^{T} \omega_t \, \mathbb{E}_{\mathbf{m}, \, \mathbf{z}_t \sim q_t(\cdot|\mathbf{x})}\big[\tilde{\mathcal{L}}_t(\theta; \mathbf{x}, a, \mathbf{m})\big] \, + \, C, \tag{7}$$

*where $C$ is independent of $\theta$. Therefore, $\mathcal{R}_{block}(\theta) \leq \mathcal{L}_{BW\text{-}SFT}(\theta) + C'$.*

*Intuition.* By focusing diffusion training on individual blocks with proper conditioning, we obtain a tractable upper bound on the exact inference objective. Minimizing this bound improves blockwise generation quality. (Proof in Appendix A.6.)

**Theorem 3.3** (Unbiased gradient of the blockwise ELBO). *Let $\tilde{\omega}$ denote the normalized weights $\tilde{\omega}_t \propto \omega_t$. Sampling block $a \sim \rho$ and diffusion step $t \sim \tilde{\omega}$ yields the stochastic gradient estimator*

$$\widehat{g}(\theta) = \frac{1}{\rho(a)} \left(\sum_{s=1}^{T} \omega_s\right) \nabla_\theta \, \mathbb{E}_{\mathbf{m}, \, \mathbf{z}_t \sim q_t(\cdot|\mathbf{x})}\big[\tilde{\mathcal{L}}_t(\theta; \mathbf{x}, a, \mathbf{m})\big]. \tag{8}$$

*Then*

$$\mathbb{E}_{\mathbf{x}, \, a \sim \rho, \, t \sim \tilde{\omega}, \, \mathbf{m}, \, \mathbf{z}_t}\big[\widehat{g}(\theta)\big] = \nabla_\theta \, \mathcal{L}_{BW\text{-}SFT}(\theta). \tag{9}$$

*Intuition.* Although each update samples only one block and one diffusion step, the resulting gradients are unbiased estimates of the surrogate objective in Eq. equation 6. This enables efficient training while retaining a clear variational connection to the true blockwise generation risk. (Proof in Appendix A.6.)

**Summary.** The blockwise objective in Eq. equation 6 bridges the gap between tractable training and faithful inference by: (i) providing a variational bound on the true generation risk, (ii) enabling unbiased stochastic optimization of the blockwise diffusion surrogate, and (iii) eliminating the systematic bias inherent in classical approaches.

## 3.4 PRACTICAL IMPLEMENTATION

We now translate the theoretical framework into a practical training algorithm. Algorithm 1 implements one training step of Blockwise SFT. The key insight is simplicity: we modify only the masking strategy, leaving the model architecture and inference procedure unchanged.

**Implementation details.** Blockwise SFT is a drop-in replacement for classical SFT: the only change is a structured mask that preserves the prefix and hides all future tokens. We keep the standard

---

**Algorithm 1** Blockwise SFT — One Training Step

---

**Require:** Instruction–response pair $\mathbf{x} = [\mathbf{c}; \mathbf{r}]$; block size $B$; diffusion steps $T$; weights $\{\omega_t\}_{t=1}^T$; model $\theta$
 1: Partition response into blocks $\mathbf{r} = [\mathbf{b}^{(1)}, \ldots, \mathbf{b}^{(M)}]$ where $M = \lceil L_r/B \rceil$
 2: **Sample active block** $a \sim \mathrm{Uniform}\{1, \ldots, M\}$
 3: Sample mask rate $\pi \sim \mathrm{Uniform}(0, 1)$
 4: **Construct training mask** $\mathbf{m}_{1:L}$:
   - Prefix ($i \leq L_c + B(a-1)$): $m_i = 0$
   - Active block ($i \in \mathcal{I}_a$): $m_i \sim \mathrm{Bernoulli}(\pi)$
   - Suffix ($i > L_c + Ba$): $m_i = 1$
 5: Sample diffusion step $t$ with probability $\propto \omega_t$
 6: Apply noise: $\mathbf{z}_t \sim q_t(\cdot \mid \mathbf{x})$
 7: Compute loss **only on active block**: $\mathcal{L} = -\sum_{i \in \mathcal{I}_a} \log p_\theta(x_i \mid \mathbf{z}_t, t)$
 8: Backpropagate through active block only; treat prefix/suffix as constants
 9: Update $\theta$ using gradient $\nabla_\theta \mathcal{L}$ with standard optimizer

---

diffusion training recipe (Appendix A.1); each step uses the stochastic gradient estimator in Theorem 3.3, with uniform block sampling $\rho(a) = 1/M$ and importance-weighted diffusion steps to optimize—without bias—the variational bound in Theorem 3.2. In practice, we use the response length $L=128$ to balance speed and context; uniform block sampling works well, though difficulty-aware reweighting could further improve efficiency. The method requires no architectural changes and integrates cleanly with existing diffusion LM codebases. By maintaining clean prefixes, hiding suffixes, and focusing loss on the active block, Blockwise SFT aligns training with blockwise inference and removes the systematic bias of classical SFT at comparable compute.

## 4 EXPERIMENTS

### 4.1 EXPERIMENT SETUP

We train on `meta-math/MetaMathQA` (Yu et al., 2024) and evaluate on the test splits of `openai/gsm8k` (Cobbe et al., 2021) and `HuggingFaceH4/MATH-500` (Hendrycks et al., 2021), reporting *Pass@1* via exact string match. These mathematical reasoning sets align with our blockwise paradigm: each step depends causally on preceding derivations without access to future tokens, mirroring clean-prefix, hidden-future supervision.

**Configurations.** We compare (1) base (no fine-tuning), (2) Classical SFT with random masking over the full response, and (3) Blockwise SFT with active-block masking. All experiments use `GSAI-ML/LLaDA-8B-Instruct` with LoRA (Hu et al., 2021), following the recipe of Biderman et al. (2024).

**Evaluation Protocols.** Recall that Classical SFT aggregates loss over the entire response each step, whereas Blockwise SFT aggregates loss only on the active block (clean prefix, fully hidden suffix). To isolate supervision granularity from raw compute, we report two comparisons: (i) hold model-side compute fixed while changing which tokens receive loss (EQUAL-FLOPS); and (ii) hold the total number of supervised tokens fixed while allowing the number of optimization steps to differ (EQUAL-TOKENS).

**EQUAL-FLOPS.** We fix the FLOP-determining knobs: parameter count, sequence length $L$, batch size, gradient accumulation, and number of optimizer updates. Thus, both methods consume identical forward and backward FLOPs; only the loss-bearing positions differ. For example, with $L_c=32$, $L_r=96$, and block size $B=32$, both run the same updates; Classical applies loss over masked positions among 96 response tokens per step, while Blockwise applies loss only within the sampled active block of 32 tokens.

**EQUAL-TOKENS.** We equalize the total number of supervised tokens over the run and index progress by $\tau$ (traversals): for Classical SFT, $\tau$ equals epochs; for Blockwise, $\tau=1$ means every sample's blocks are each supervised once (requiring $\approx L_r/B$ epochs). In the same example, Classi-

Table 1: Head-to-head comparison under EQUAL-FLOPS: Pass@1 (%) on GSM8K and MATH.

| Method | GSM8K (Pass@1) | MATH (Pass@1) |
|---|---|---|
| **Base (no FT)** | $62.1 \pm 1.0$ | $31.7 \pm 0.4$ |
| **Classical SFT** (Austin et al., 2023) | $67.7 \pm 1.4$ | $29.6 \pm 0.7$ |
| MDLM (Sahoo et al., 2024) | $68.4 \pm 1.9$ | $31.9 \pm 0.5$ |
| Soft-Masked Diffusion LM (Chen et al., 2023) | $67.7 \pm 0.8$ | $29.9 \pm 0.6$ |
| RDM (Zheng et al., 2024) | $65.5 \pm 1.5$ | $32.3 \pm 0.6$ |
| Two-Step Loss & Scheduling (Asada & Miwa, 2025) | $70.8 \pm 1.3$ | $32.6 \pm 0.7$ |
| **Blockwise SFT (Ours)** | $\mathbf{76.0 \pm 1.6}$ | $\mathbf{34.2 \pm 0.5}$ |

cal supervises 96 tokens/step while Blockwise supervises 32; accordingly, Blockwise runs $3\times$ more epochs to reach the same $\tau$.

**Implementation Details.** Bernoulli masking rate $\pi \sim \text{Uniform}(10^{-3}, 1)$ is sampled per sequence (Classical) or per active block (Blockwise); if nothing is masked, we mask the last token. Training uses AdamW (Loshchilov & Hutter, 2019) with linear warmup and cosine decay (Smith, 2017) in `bfloat16` (Micikevicius et al., 2018), runs on a single A100 (80GB) at $\sim$5.1k tokens/s, and peaks at $\approx$23.5GB memory (deployable on RTX 3090) when the batch size is set to 1. Full hyperparameters are in Appendix A.1.

## 4.2 MAIN RESULTS

Table 1 presents our primary results comparing Blockwise SFT against the original model and five baseline methods under matched compute. Blockwise SFT achieves the highest Pass@1 accuracy on both benchmarks, with a substantial +5.2 point gain over the strongest baseline (Two-Step Loss & Scheduling) on GSM8K and +1.6 points on MATH. Notably, Classical SFT underperforms the base model on MATH (29.6 vs. 31.7), suggesting that naive supervision misalignment can be detrimental. Recent variants offer only marginal improvements, while Blockwise SFT's aligned supervision delivers consistent and significant gains.

**Training Dynamics under EQUAL-FLOPS and EQUAL-TOKENS.** Figs. 3a–b show that Blockwise SFT converges to a lower loss (0.24) than Classical SFT (0.36), indicating more effective learning under aligned supervision. Figs. 4a–b track test Pass@1 throughout training: on GSM8K, Blockwise SFT maintains a $\sim$10-point margin; on MATH, Classical SFT degrades below the base model while Blockwise SFT improves steadily. These dynamics highlight that training–inference alignment is crucial to preserve and enhance reasoning ability.

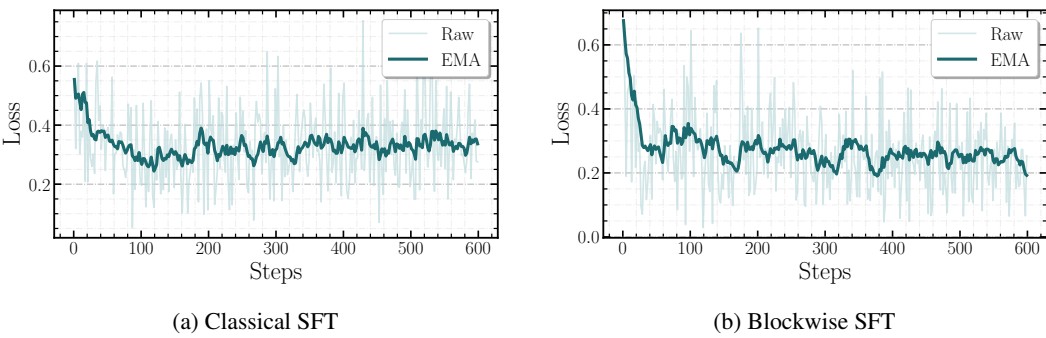

(a) Classical SFT                    (b) Blockwise SFT

Figure 3: Training loss on MetaMathQA under EQUAL-FLOPS.

Fig. 5 evaluates accuracy under a fixed supervised-token budget. Blockwise SFT delivers large gains at $\tau$=1 and remains stable through $\tau$=3. By contrast, Classical SFT behaves differently across datasets: on GSM8K it shows a small early uptick at $\tau$=1 before degrading as $\tau$ increases, whereas on MATH it declines monotonically across traversals. The late-epoch drops further indicate clear

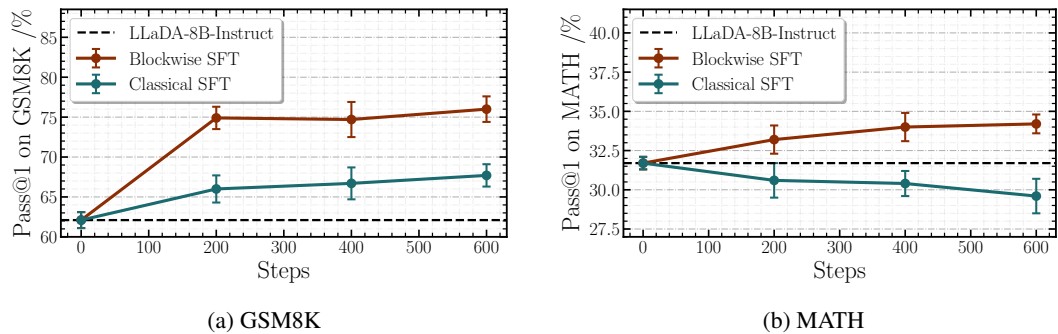

(a) GSM8K

(b) MATH

Figure 4: Test Pass@1 during training under EQUAL-FLOPS.

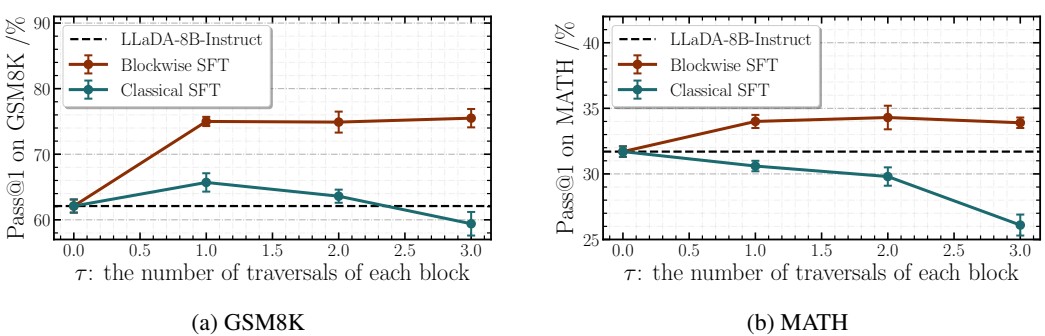

(a) GSM8K

(b) MATH

Figure 5: Test Pass@1 during training under EQUAL-TOKENS.

overfitting for Classical SFT when the same tokens are revisited over multiple epochs, while Block-wise SFT remains robust under the same budget. Thus, Blockwise SFT's advantage stems from supervision quality, not quantity.

### 4.3 BLOCK SIZE CONSISTENCY STUDY

Figure 6 investigates robustness to block size misalignment between training and inference. For each dataset, we train four checkpoints with $B_{\text{train}} \in \{8, 16, 32, 64\}$ under the EQUAL-FLOPS protocol (identical sequence length, batch size, and number of optimizer updates). Each checkpoint is then evaluated with semi-autoregressive decoding at $B_{\text{infer}} \in \{8, 16, 32, 64\}$, producing a $4 \times 4$ grid per dataset. Diagonal cells capture matched granularity; off-diagonal cells impose a controlled mismatch at inference.

In both experimental groups, performance peaks along the diagonal ($B_{\text{train}} = B_{\text{infer}}$) and degrades progressively as the mismatch grows. This diagonal dominance indicates that gains stem from aligning supervision granularity with the decoding procedure, rather than from any single block size. Near-diagonal cells degrade gracefully, suggesting robustness to minor misalignment. The factorial design removes compute confounds, attributing the observed pattern to training–inference alignment.

### 4.4 ABLATION STUDIES

**Prefix Noise Analysis.** Figs. 7a–b examine the effect of corrupting prefix blocks during training. Performance degrades progressively as $\pi_{\text{prefix}}$ increases, with the steepest drop at $\pi_{\text{prefix}} = 1.0$. This degradation pattern has an important nuance: when block sizes are large, noisy prefixes often have no effect because the first block (which has no prefix) is selected more frequently. This explains why Blockwise SFT maintains reasonable performance even with substantial prefix corruption, because the fast convergence compensates for the effectively reduced training data. The key insight is that maintaining clean prefix context during training mirrors the deterministic prefix available at

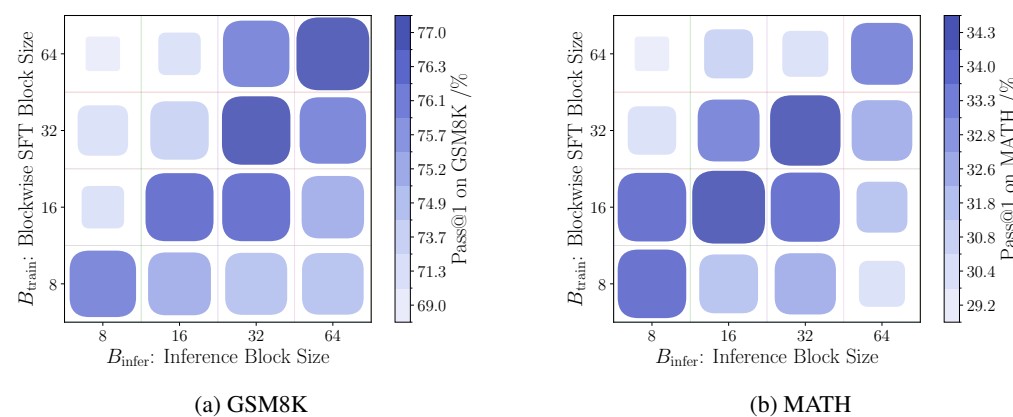

Figure 6: Test Pass@1 across $B_{\text{train}}$ and $B_{\text{infer}}$ combinations on two datasets.

inference, reinforcing proper conditioning behavior. Implementation details are provided in Appendix A.7.

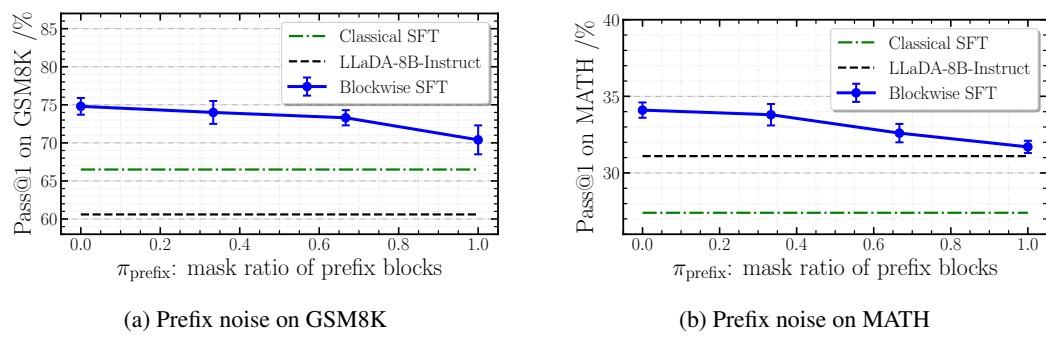

Figure 7: Prefix noise ablation under Blockwise SFT: (a) GSM8K and (b) MATH. Performance consistently degrades as $\pi_{\text{prefix}}$ increases, highlighting the importance of clean prefixes.

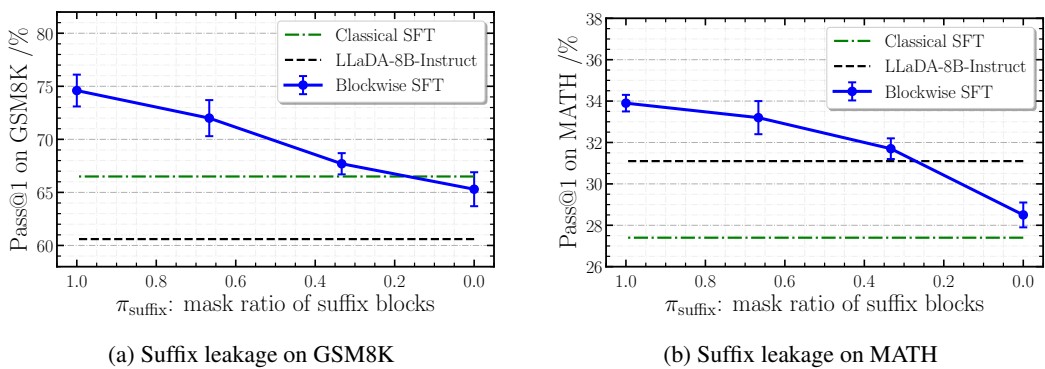

Figure 8: Suffix leakage ablation under Blockwise SFT: (a) GSM8K and (b) MATH. Making future tokens visible (smaller $\pi_{\text{suffix}}$) sharply harms performance, underscoring the need for strict future masking.

**Suffix Leakage Analysis.** Figs. 8a–b reveal that suffix visibility is critically detrimental. As $\pi_{\text{suffix}}$ decreases (more future tokens become visible), performance drops sharply. At $\pi_{\text{suffix}} = 0$ (fully visible suffix), Blockwise SFT degenerates to Classical SFT performance. This dramatic degradation occurs because visible suffixes fundamentally corrupt the learning objective: instead of learning causal reasoning from prefix to active block, the model learns to extract answers from future context,

which is a capability unavailable at inference. This finding underscores that strict future masking is the primary mechanism behind Blockwise SFT's success, ensuring the model develops genuine reasoning capabilities rather than spurious correlations with leaked information.

## 5 CONCLUSION

We introduce Blockwise SFT, a simple recipe that aligns supervision with semi-autoregressive, blockwise decoding in discrete diffusion LMs. By freezing clean prefixes, masking only the active block, and strictly hiding future tokens, it removes the contextual shift and dependency leakage of classical SFT. We ground this method with a variational upper bound on blockwise likelihoods, unbiased block/time-sampled gradients, and an analysis of classical SFT's mismatch bias (Theorems 3.1–3.2). Empirically, fine-tuning on MetaMathQA with evaluation on GSM8K and MATH yields consistent gains under matched compute and token budgets, state-of-the-art head-to-head results, and diagnostics that track alignment (block-size consistency in §4.3, prefix/suffix ablations in §4.4). The recipe is easy to adopt: no architectural changes, a drop-in loss, and practical single-step estimators (§3.4). Limitations and next steps include adaptive or uncertainty-aware block sizing, combining with preference or RL-style objectives, and extending the alignment principle to other semi- or non-autoregressive generators. Overall, honoring the decoder's structural constraints in SFT is a core driver of performance in diffusion-based language models.

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

# A APPENDIX

## A.1 FULL HYPERPARAMETER SETTINGS

Table 2 lists the full hyperparameter configuration used in all experiments reported in the main paper. Unless otherwise specified, parameters not listed (e.g., weight decay, gradient clipping) are set to their default values. The configuration is kept identical across all experimental protocols (EQUAL-FLOPS, EQUAL-TOKENS) and other studies to ensure fair comparison.

Table 2: hyperparameter configuration for all experiments.

| Category | Parameter | Value / Description |
|---|---|---|
| **Batch Size** | Number of GPUs | 1 |
| | Per-device batch size | 32 |
| | Gradient accumulation steps | 1 |
| | Global batch size | 32 |
| **Learning Rate Schedule** | Learning rate | $1 \times 10^{-5}$ |
| | Scheduler type | Warmup + Cosine decay |
| | Warmup steps | 10% of total steps |
| | Cosine decay minimum ratio | 0.1 |
| **Masking** | Mask ratio $\pi$ | Uniform$(10^{-3}, 1)$ sampled per sequence / active block |
| **Optimizer (AdamW)** | $\beta_1$ | 0.95 |
| | $\beta_2$ | 0.99 |
| | Weight decay | 0 |
| **LoRA** | Rank $r$ | 256 |
| | $\alpha$ | 512 |
| | Dropout | 0.05 |
| | Target modules | Attention and MLP layers |
| | Bias | None |

## A.2 DATASET PREPROCESSING AND EVALUATION PROTOCOLS

**Training Set.** We follow the preprocessing instructions provided by the MetaMathQA dataset. For each sample, the `query` field is inserted into the following instruction template:

```
Below is an instruction that describes a task.
Write a response that appropriately completes the request.

### Instruction:
{instruction}

### Response: Let's think step by step.
```

The placeholder {instruction} is replaced with the actual `query` text. We then concatenate this instruction with the `response` field using the model's chat template to form the final training sequence. To reduce compute and facilitate reproducibility, the maximum sequence length is set to 256 tokens; samples exceeding this limit (fewer than 10% of the dataset) are discarded. The `MetaMathQA` dataset contains no samples from the GSM8K or MATH test sets.

**Test Sets.** For evaluation, we apply the same instruction template to the prompts from the test splits of GSM8K and MATH(MATH-500), ensuring identical formatting between training and evaluation inputs.

**Scoring Rules.** We report Pass@1 accuracy for both datasets using exact match:

- **GSM8K:** We extract the last numeric value from the model output (e.g., using a regex that matches the final signed/decimal number) and perform exact string match against the reference answer.
- **MATH:** No extraction or normalization is applied because of the complexity of the answer format; we compare the full model output to the reference answer by exact string match.

## A.3 DETAILED INFERENCE SETTINGS

Table 3 summarises the inference-time configurations used throughout our experiments, including both the default settings and the variations for the Block Size Consistency Study. Unless otherwise noted, all other parameters follow the default configuration of the `LLaDA-8B-Instruct` model.

Table 3: Inference-time settings for all experiments.

| Parameter | Value / Range | Description |
|---|---|---|
| Maximum new tokens | 128 | Maximum number of tokens generated per example. |
| Block size | $\{8, 16, 32, 64\}$
8
32 (default) | Block Size Consistency Study.
Ablation experiments.
Default for all other experiments. |
| Diffusion steps | 128 (default) | Number of denoising steps during inference. |
| Other parameters | Default (`LLaDA-8B-Instruct`) | All remaining inference parameters follow the model's official default configuration. |

## A.4 ADDITIONAL PRELIMINARIES AND NOTATION

**Tokenisation and sequences.** Let $\mathcal{V}$ be a finite vocabulary of size $V$. A tokenised sequence is $\mathbf{x}_{1:L} \in \mathcal{V}^L$ with $x_i \in \mathcal{V}$. Each training instance is

$$\mathbf{x}_{1:L} = \left[ \mathbf{c}_{1:L_c}; \mathbf{r}_{1:L_r} \right], \qquad L = L_c + L_r.$$

**Masking operators.** Let $\mathbf{m}_{1:L} \in \{0,1\}^L$ with $m_i=0$ for observed and $m_i=1$ for masked. For **Classical SFT** we sample i.i.d. masks on all response positions: $m_i \sim \text{Bernoulli}(\pi)$ for $i \in \{L_c+1, \dots, L\}$. For **Blockwise SFT**, with active block index $a$ and block size $B$,

$$m_i = \begin{cases} 0, & i \le L_c + B(a-1) \quad \text{(prefix)}, \\ \text{Bernoulli}(\pi), & L_c + B(a-1) < i \le L_c + Ba \quad \text{(active block)}, \\ 1, & i > L_c + Ba \quad \text{(suffix)}. \end{cases}$$

We use $\tilde{\mathbf{x}} = \mathbf{x} \odot (1 - \mathbf{m})$ to denote the masked sequence. (For variants used in ablations, see Eq. equation 11 in Appendix A.7.)

**Discrete diffusion objective.** Let $t \in \{0, \dots, T\}$ index diffusion steps and $q_t(\mathbf{z}_t \mid \mathbf{x})$ be the forward noising kernel (relaxed categorical diffusion). The reverse model $\mathbf{p}_\theta(\mathbf{x} \mid \mathbf{z}_t, t)$ predicts the original tokens at masked positions. The per–step cross-entropy and the weighted objective are

$$\mathcal{L}_t(\theta) = \mathbb{E}_{\mathbf{x}, \mathbf{m}, \mathbf{z}_t} \left[ \text{CE} \big( \mathbf{p}_\theta(\cdot \mid \mathbf{z}_t, t), \mathbf{x} \big) \mid \mathbf{m} \right], \qquad \sum_{t=1}^{T} \omega_t \, \mathcal{L}_t(\theta),$$

where $\{\omega_t\}_{t=1}^{T}$ are nonnegative weights.

**Notation table.** See Table 4.

| Symbol | Meaning |
|---|---|
| $\mathcal{V}, V$ | Vocabulary and its size |
| $\mathbf{x}_{1:L}$ | Full instruction–response sequence |
| $L_c, L_r$ | Instruction and response lengths |
| $B, M$ | Block size and number of response blocks |
| $a$ | Index of the active block |
| $\mathbf{m}_{1:L}$ | Binary mask (1 = latent, 0 = observed) |
| $T, t$ | Total diffusion steps and current step |
| $q_t, \mathbf{p}_\theta$ | Forward kernel and reverse model |
| $\pi$ | Masking rate (sequence or active-block level) |

Table 4: Complete notation used in the paper.

### A.5 PRACTICAL ESTIMATORS AND WEIGHTING

**Single-$t$ sampling.** To reduce cost, we may sample a single diffusion step $t$ with probability $\tilde{\omega}_t \propto \omega_t$ and multiply the per-step gradient by the normalizer $Z = \sum_{s=1}^{T} \omega_s$. Formally, letting

$$\widehat{\mathcal{L}}(\theta; \mathbf{x}, a, t) = Z \, \mathbb{E}_{\mathbf{z}_t \sim q_t(\cdot|\mathbf{x})}\big[\tilde{\mathcal{L}}_t(\theta; \mathbf{x}, a)\big] \quad \text{with} \quad t \sim \tilde{\omega},$$

we have $\nabla_\theta \mathbb{E}_{t \sim \tilde{\omega}}[\widehat{\mathcal{L}}(\theta; \mathbf{x}, a, t)] = \nabla_\theta \sum_{t=1}^{T} \omega_t \, \mathbb{E}_{\mathbf{z}_t}[\tilde{\mathcal{L}}_t(\theta; \mathbf{x}, a)]$.

**Block sampling and importance weights.** If the active block is drawn from a non-uniform distribution $\rho(a)$, multiply the per-example gradient by $1/\rho(a)$ to obtain an unbiased estimator of the block-averaged objective (cf. Theorem 3.3).

### A.6 PROOFS FOR SECTION 3.2 AND SECTION 3.3

*Proof of Theorem 3.1.* Write the classical-SFT gradient for block $a$ as an expectation over prefix/suffix masking patterns:

$$\mathbb{E}\big[\nabla_\theta \ell^{\mathrm{cls}}(\theta; \mathbf{x}, a)\big] = \mathbb{E}_{\mathbf{m}}\big[\nabla_\theta \ell^{\mathrm{cls}}(\theta; \mathbf{x}, a; \mathbf{m})\big],$$

where $\mathbf{m}$ collects all prefix and suffix mask bits. Let $\mathbf{m}^\star$ denote the ideal mask that keeps the entire prefix clean and hides the entire suffix, so that $\nabla_\theta \ell^\star(\theta; \mathbf{x}, a) = \nabla_\theta \ell^{\mathrm{cls}}(\theta; \mathbf{x}, a; \mathbf{m}^\star)$. Then

$$\big\| \mathbb{E}\big[\nabla_\theta \ell^{\mathrm{cls}}(\theta; \mathbf{x}, a)\big] - \nabla_\theta \ell^\star(\theta; \mathbf{x}, a) \big\| = \left\| \mathbb{E}_{\mathbf{m}}\big[\nabla_\theta \ell^{\mathrm{cls}}(\theta; \mathbf{x}, a; \mathbf{m}) - \nabla_\theta \ell^{\mathrm{cls}}(\theta; \mathbf{x}, a; \mathbf{m}^\star)\big] \right\|$$

$$\leq \mathbb{E}_{\mathbf{m}}\Big[\big\|\nabla_\theta \ell^{\mathrm{cls}}(\theta; \mathbf{x}, a; \mathbf{m}) - \nabla_\theta \ell^{\mathrm{cls}}(\theta; \mathbf{x}, a; \mathbf{m}^\star)\big\|\Big],$$

where we used Jensen's inequality.

Connect $\mathbf{m}^\star$ to $\mathbf{m}$ by flipping one mask bit at a time. For each prefix token $i \in \mathcal{I}_{\mathrm{prefix}}^{(a)}$, let $\mathbf{m}^{(i)}$ and $\mathbf{m}^{(i-1)}$ be two consecutive patterns that differ only in the $i$-th prefix bit. By the bounded-difference assumption,

$$\big\|\nabla_\theta \ell^{\mathrm{cls}}(\theta; \mathbf{x}, a; \mathbf{m}^{(i)}) - \nabla_\theta \ell^{\mathrm{cls}}(\theta; \mathbf{x}, a; \mathbf{m}^{(i-1)})\big\| \leq L_{\mathrm{pre}}$$

whenever the $i$-th prefix bit is flipped. Summing along the path from $\mathbf{m}^\star$ to $\mathbf{m}$ and applying the triangle inequality give

$$\big\|\nabla_\theta \ell^{\mathrm{cls}}(\theta; \mathbf{x}, a; \mathbf{m}) - \nabla_\theta \ell^{\mathrm{cls}}(\theta; \mathbf{x}, a; \mathbf{m}^\star)\big\| \leq L_{\mathrm{pre}} \cdot N_{\mathrm{pre}}(\mathbf{m}) + L_{\mathrm{suf}} \cdot N_{\mathrm{suf}}(\mathbf{m}),$$

where $N_{\mathrm{pre}}(\mathbf{m})$ is the number of prefix positions whose mask differs from the ideal pattern, and $N_{\mathrm{suf}}(\mathbf{m})$ is defined analogously for suffix positions.

Taking expectation over $\mathbf{m}$ yields

$$\big\| \mathbb{E}\big[\nabla_\theta \ell^{\mathrm{cls}}(\theta; \mathbf{x}, a)\big] - \nabla_\theta \ell^\star(\theta; \mathbf{x}, a) \big\| \leq L_{\mathrm{pre}} \, \mathbb{E}\big[N_{\mathrm{pre}}(\mathbf{m})\big] + L_{\mathrm{suf}} \, \mathbb{E}\big[N_{\mathrm{suf}}(\mathbf{m})\big].$$

Under classical SFT, each prefix token is independently corrupted with probability $\pi$, so $\mathbb{E}[N_{\text{pre}}(\mathbf{m})] = \pi |\mathcal{I}_{\text{prefix}}^{(a)}|$. Each suffix token is independently left visible with probability $(1 - \pi)$, so $\mathbb{E}[N_{\text{suf}}(\mathbf{m})] = (1 - \pi) |\mathcal{I}_{\text{suffix}}^{(a)}|$. Substituting these expectations gives

$$\left\| \mathbb{E}\big[\nabla_\theta \ell^{\text{cls}}(\theta; \mathbf{x}, a)\big] - \nabla_\theta \ell^\star(\theta; \mathbf{x}, a) \right\| \le L_{\text{pre}} \cdot \pi \left| \mathcal{I}_{\text{prefix}}^{(a)} \right| + L_{\text{suf}} \cdot (1 - \pi) \left| \mathcal{I}_{\text{suffix}}^{(a)} \right|,$$

which is Eq. equation 2. Averaging over $a$ extends the result to the population gradient bias. $\qquad\square$

*Proof of Theorem 3.2.* Consider the joint over $(\mathbf{b}^{(a)}, \{\mathbf{z}_t\}_{t=1}^T)$ under the forward noising process $q_t(\cdot \mid \mathbf{x})$ restricted to indices $\mathcal{I}_a$, conditioning on $\mathbf{context}^{(a)}$ and deterministically masking tokens outside $\mathcal{I}_a$ (suffix). The discrete-diffusion ELBO applied to this restricted chain gives

$$-\log p_\theta\big(\mathbf{b}^{(a)} \mid \mathbf{context}^{(a)}\big) \le \sum_{t=1}^T \mathbb{E}_{q_t}\Big[\text{KL}\big(q(\mathbf{z}_{t-1} \mid \mathbf{z}_t, \mathbf{b}^{(a)}) \,\|\, p_\theta(\mathbf{z}_{t-1} \mid \mathbf{z}_t, \mathbf{context}^{(a)})\big)$$
$$- \log p_\theta\big(\mathbf{b}^{(a)} \mid \mathbf{z}_1, \mathbf{context}^{(a)}\big)\Big] + C. \tag{10}$$

Standard manipulations for discrete diffusion convert the KL and reconstruction terms into token-wise cross-entropies on $\mathcal{I}_a$ under intra-block masking. Taking the additional expectation over the intra-block mask $\mathbf{m}$ and collecting constants yields a weighted sum

$$\sum_{t=1}^T \omega_t \, \mathbb{E}_{\mathbf{m}, \, \mathbf{z}_t \sim q_t(\cdot | \mathbf{x})} \big[\tilde{\mathcal{L}}_t(\theta; \mathbf{x}, a, \mathbf{m})\big],$$

where $C$ and the weights $\{\omega_t\}$ depend only on the forward process and are independent of $\theta$. Taking expectation over data and summing across $a$ yields $\mathcal{R}_{\text{block}}(\theta) \le \mathcal{L}_{\text{BW–SFT}}(\theta) + C'$ with $C'$ independent of $\theta$. $\qquad\square$

*Proof of Theorem 3.3.* Taking total expectation over $\mathbf{x}$, $a \sim \rho$, $t \sim \tilde{\omega}$, and over the stochastic mask $\mathbf{m}$ and noise $\mathbf{z}_t$ used to evaluate $\tilde{\mathcal{L}}_t$, we obtain

$$\mathbb{E}\big[\widehat{g}(\theta)\big] = \sum_{a=1}^M \rho(a) \frac{1}{\rho(a)} \sum_{t=1}^T \tilde{\omega}_t \Big(\sum_{s=1}^T \omega_s\Big) \nabla_\theta \, \mathbb{E}_{\mathbf{m}, \, \mathbf{z}_t} \big[\tilde{\mathcal{L}}_t(\theta; \mathbf{x}, a, \mathbf{m})\big]$$

$$= \sum_{a=1}^M \sum_{t=1}^T \omega_t \nabla_\theta \, \mathbb{E}_{\mathbf{m}, \, \mathbf{z}_t} \big[\tilde{\mathcal{L}}_t(\theta; \mathbf{x}, a, \mathbf{m})\big]$$

$$= \nabla_\theta \, \mathbb{E}_{\mathbf{x}, \, a} \sum_{t=1}^T \omega_t \mathbb{E}_{\mathbf{m}, \, \mathbf{z}_t} \big[\tilde{\mathcal{L}}_t(\theta; \mathbf{x}, a, \mathbf{m})\big]$$

$$= \nabla_\theta \, \mathcal{L}_{\text{BW–SFT}}(\theta),$$

where we used the definition of $\tilde{\omega}_t \propto \omega_t$ to simplify the weighted sum over $t$ and the fact that differentiation commutes with finite expectation. This shows that $\widehat{g}(\theta)$ is an unbiased estimator of the gradient of the blockwise diffusion surrogate in Eq. equation 6. $\qquad\square$

## A.7 ABLATION IMPLEMENTATION DETAILS

We formalize the two ablation settings (Noisy Prefix and Leaky Suffix) as direct modifications to the masking rule in Blockwise SFT (§3.4). Recall that in standard Blockwise SFT, the binary mask $m_i$ for token $i$ is defined as:

$$m_i = \begin{cases} 0, & i \le L_c + B(a-1) \quad \text{(prefix)}, \\ \text{Bernoulli}(\pi), & L_c + B(a-1) < i \le L_c + Ba \quad \text{(active block)}, \\ 1, & i > L_c + Ba \quad \text{(suffix)}, \end{cases} \tag{11}$$

where $\pi \in (0, 1)$ is the active-block masking rate, fixed for a given sample. Let $\mathcal{I}_{\text{prefix}} = \{L_c + 1, \dots, L_c + B(a-1)\}$ denote all prefix tokens (after the prompt and before the active block), $\mathcal{I}_a$ the indices of the active block, and $\mathcal{I}_{\text{suffix}} = \{L_c + Ba + 1, \dots, L\}$ the suffix tokens.

**1. Noisy Prefix.** Given a fixed $\pi_{\text{prefix}} \in [0, 1]$ for the entire experiment, the mask $m_i$ is modified from equation 11 as:

$$m_i = \begin{cases} \text{Bernoulli}(\pi_{\text{prefix}}), & i \in \mathcal{I}_{\text{prefix}}, \\ \text{Bernoulli}(\pi), & i \in \mathcal{I}_a, \\ 1, & i \in \mathcal{I}_{\text{suffix}}, \end{cases}$$

injecting random noise into the prefix tokens and simulating contextual distribution shift.

**2. Leaky Suffix.** Given a fixed $\pi_{\text{suffix}} \in [0, 1]$ for the entire experiment, the mask $m_i$ is modified from equation 11 as:

$$m_i = \begin{cases} 0, & i \in \mathcal{I}_{\text{prefix}}, \\ \text{Bernoulli}(\pi), & i \in \mathcal{I}_a, \\ \text{Bernoulli}(\pi_{\text{suffix}}), & i \in \mathcal{I}_{\text{suffix}}, \end{cases}$$

allowing the model to observe a random fraction of future tokens beyond the active block, thus introducing block-level dependency leakage.

In both cases, $\pi_{\text{prefix}}$ and $\pi_{\text{suffix}}$ are fixed for the duration of an experiment and applied identically to all training samples.

## A.8 DISCLOSURE OF LARGE LANGUAGE MODEL USAGE

Large Language Models were used exclusively for grammatical refinement and language polishing of this manuscript. All research design, methodology, experiments, analysis, and scientific conclusions are original contributions. The LLMs provided no substantive input on technical content, experimental design, or theoretical derivations.

