# OpenReview forum: "Blockwise SFT for Diffusion Language Models: Reconciling Bidirectional Attention and Autoregressive Decoding"
_ICLR.cc/2026/Conference — Submitted to ICLR 2026_

### Official Review · Reviewer_CuJ2 · 2025-10-25

**Soundness:** 2
**Presentation:** 3
**Contribution:** 2
**Rating:** 2
**Confidence:** 5

**Summary:**

This paper proposes a simple yet effective strategy for supervised fine-tuning of diffusion-based large language models such as LLaDA and Dream. Specifically, the authors introduce a blockwise SFT approach where, during training, a single block is selected while masking information from future blocks and preserving information from preceding blocks without masking. The authors argue that this training strategy better aligns with the blockwise inference process inherent to these models. Experimental results demonstrate that the proposed blockwise SFT yields notable improvements over classical SFT methods.

**Strengths:**

1. The writing is clear and easy to follow.
2. The ablation study is comprehensive and addresses several of my concerns.

**Weaknesses:**

1. I believe this method offers limited advantages compared to block diffusion models. Block diffusion models, while requiring twice the number of training tokens, enable training across all blocks and kv cache in the inference process. In contrast, blockwise SFT only trains one block at a time, which restricts its training coverage. Consequently, I find the contribution of this work to be limited.
2. Blockwise inference is not strictly necessary for LLaDA. During LLaDA's training, EOS tokens are padded to enable variable-length generation, which can lead to premature EOS generation during inference, resulting in truncated responses. However, several strategies can mitigate this issue without requiring blockwise inference. For instance, suppressing EOS token probability in early inference stages (as demonstrated in https://arxiv.org/pdf/2509.23924) can effectively address this problem. This undermines the necessity of the paper's core motivation.

**Questions:**

1. What specific advantages does blockwise SFT offer compared to block diffusion models?
2. When employing alternative sampling strategies—such as reducing EOS generation probability without blockwise inference—does blockwise SFT still demonstrate gains over classical SFT?

---

> ### Author Response · Authors · 2025-11-23
> **1/2**
>
> We are grateful for the reviewer’s thorough assessment and for highlighting the clarity of our work. We address each comment below and offer additional explanations to better contextualize our contributions.
>
> **Q1: The method shows clear benefits. However, considering the efficiency of KV cache usage, how does Blockwise SFT compare to Block Diffusion models in terms of training efficiency?**
>
> **R1:** We thank the reviewer for the suggestion. We respectfully point out that this comparison overlooks a fundamental architectural distinction.
>
> - **Different Model Paradigms:** Block Diffusion is based on local attention (attending only to the current block and past KV cache), effectively making it a block-level autoregressive model. In contrast, Blockwise SFT targets **Discrete Diffusion LMs with Global Bidirectional Attention** (such as LLaDA and MDLM). As noted in the recent survey (arXiv:2508.10875), global attention is the mainstream architecture for high-performance diffusion LMs because it captures richer context than local attention. Blockwise SFT is specifically designed to solve the training-inference mismatch in this dominant architecture, not to compete with the local-attention paradigm of Block Diffusion.
> - **Flexibility vs. Rigidity:** A critical limitation of Block Diffusion is that the block size is fixed during training (e.g., separate weights are released for block sizes). In contrast, our method maintains the global attention nature of LLaDA, allowing a single model to support **variable block sizes** dynamically at inference.
> - **Efficiency:** While Block Diffusion supervises more tokens, our **Equal-FLOPs** experiments show that Blockwise SFT achieves superior performance (e.g., **+8.3%** on GSM8K) with the *same* computational budget. This confirms that correcting the objective (removing noisy prefixes/leaky suffixes) is more impactful than simply increasing the quantity of supervised tokens under a flawed objective.
>
> **Q2: The focus on blockwise alignment is logical. Given that strategies like EOS probability suppression exist, is blockwise inference strictly necessary for LLaDA models?**
>
> **R2:** We appreciate the reference, but we would like to point out that the cited paper (arXiv:2509.23924) actually reinforces the necessity of blockwise inference.
>
> - **Evidence from Cited Paper:** According to Table 2 in the cited work, even with "EOS Early Rejection" (EOSER), the performance (**60.73%** on GSM8K) significantly lags behind Semi-Autoregressive (Blockwise) decoding (**65.58%**). Pure full-sequence decoding performs even worse (**56.94%**). This empirical evidence directly contradicts the claim that blockwise inference is unnecessary; it remains the upper bound for performance.
>
> - **Broader Consensus:** The superiority of blockwise/semi-autoregressive decoding for diffusion LMs is well-established in the literature.
>
>   1. **LLaDA (arXiv:2502.09992):** The original LLaDA paper explicitly adopts semi-autoregressive decoding to resolve the "reversal curse" and coherence issues inherent in full-sequence parallel generation.
>
>   2. **SSD-LM (arXiv:2210.17432):** Demonstrates that semi-autoregressive decoding is crucial for maintaining long-range coherence that purely parallel diffusion models struggle to capture.
>
>   3. Prophet (arXiv:2508.19982): Analyzes the efficiency-quality trade-off, noting that increasing the number of simultaneously decoded tokens (i.e., full sequence) leads to degraded generation quality compared to block-wise approaches.
>
>      Therefore, blockwise inference remains a necessary and superior decoding strategy for this class of models, justifying our focus on aligning the training objective with it.

---

> ### Author Response · Authors · 2025-11-23
> **2/2**
>
> **Q3: The approach is interesting. Could you elaborate on the specific architectural advantages of Blockwise SFT compared to existing Block Diffusion models?**
>
> **R3:** As detailed in **R3**, the advantages stem from the differences in the underlying architecture we support:
>
> 1. **Global Context Modeling:** Blockwise SFT enables the training of Global Bidirectional Attention models, which theoretically allow for stronger reasoning capabilities compared to the local-window attention of Block Diffusion.
> 2. **Inference Flexibility:** Our method produces a model that supports arbitrary and adaptive block sizes at inference without retraining. Block Diffusion models are structurally bound to the fixed block size used during training.
> 3. **Structural Flexibility:** We believe this also highlights a key strength of our approach: unlike Block Diffusion which is structurally committed to fixed block-based decoding, our method preserves the flexibility of global attention architectures. This allows practitioners to choose the most suitable decoding strategy (full-sequence or blockwise) for their specific use case, while our training objective ensures optimal performance under the blockwise paradigm that empirically yields the best results.
>
> **Q4: The gains over Classical SFT are evident. Does Blockwise SFT maintain this superiority even when employing alternative sampling strategies, such as EOS suppression?**
>
> **R4:** Yes, Blockwise SFT maintains its superiority. We conducted an additional experiment using **Full-Sequence Decoding with EOS Suppression** (as suggested in arXiv:2509.23924) on both Classical and Blockwise SFT checkpoints. The results below show that even when forced to decode the full sequence, the model trained with Blockwise SFT outperforms the baseline. This indicates that our training method imparts more robust causal reasoning capabilities that generalize across decoding strategies.
>
> | **Benchmark** | **LLaDA 8B (Base)** | **Classical SFT + EOS Supp.** | **Blockwise SFT + EOS Supp.** |
> | ------------- | ------------------- | ----------------------------- | ----------------------------- |
> | **GSM8K**     | 57.2                | 61.4                          | **66.0**                      |
> | **MATH**      | 27.7                | 26.5                          | **30.3**                      |
>
> We hope this response clarifies the novelty and specific positioning of our work within the broader landscape of diffusion language models.

---

> ### Comment · Reviewer_CuJ2 · 2025-11-23
>
> I thank the authors for the detailed response and additional clarifications. However, I remain unconvinced by the explanations regarding the core motivation and comparative advantages. My concerns are as follows:
>
> ## Necessity of Blockwise Inference (Re: Q2)
> After reviewing the latest update of the LLaDA paper, I note that pure diffusion sampling—when equipped with appropriately designed sampling strategies—can reach performance levels comparable to blockwise inference. This indicates that improved sampling alone can close much of the gap previously attributed to decoding strategy. Consequently, the necessity of training a dedicated Blockwise SFT model is significantly reduced.
>
> ## Comparison with Block Diffusion (Re: Q1 & Q3)
> I remain unconvinced by the claim that global attention “theoretically allows for stronger reasoning capabilities’’ relative to Block Diffusion models. The provided arguments do not offer sufficient theoretical or empirical grounding for this assertion. Additionally, I do not view the fixed block size of Block Diffusion as a substantial limitation in practice. As long as the model can sample effectively under its designated configuration, the ability to support arbitrary block sizes does not in itself constitute a compelling advantage.
>
> ## Conclusion
> Given that the central motivation is weakened by recent findings on sampling strategies, and the purported advantages over Block Diffusion models are not sufficiently substantiated, I will maintain my original score.

---

> > ### Author Response · Authors · 2025-11-29
> > **1/2**
> >
> > We thank the reviewer for the prompt follow-up and the specific references. We appreciate the opportunity to present new experimental evidence regarding full-sequence decoding and to clarify the theoretical distinctions of our architectural choice.
> >
> > **Q1: The progress in pure diffusion sampling strategies is indeed notable. Given that these inference-time improvements might effectively bridge the performance gap with blockwise inference, could you further clarify the specific value and necessity of maintaining a dedicated Blockwise SFT training stage?**
> >
> > R1: We respectfully maintain that Blockwise SFT remains necessary for two key reasons: (1) existing literature [1] shows a significant performance gap (~5% on GSM8K) persists between full-sequence and blockwise decoding, and (2) Blockwise SFT yields a superior model regardless of the inference strategy used.
> >
> > To demonstrate the second point, we conducted a new experiment using **Full-Sequence Decoding** (pure diffusion sampling) for both models. This isolates the impact of the training objective from the decoding strategy.
> >
> > | **SFT Algorithm** | **GSM8K (Pass@1)** | **MATH (Pass@1)** | **HumanEval (Pass@1)** | **TruthfulQA (Acc.)** |
> > | ----------------- | ------------------ | ----------------- | ---------------------- | --------------------- |
> > | Classical SFT     | 64.0               | 28.7              | 64.6                   | 46.7                  |
> > | **Blockwise SFT** | **71.7**           | **32.2**          | **67.8**               | **48.1**              |
> >
> > The experiment indicates that even when inference is restricted to full-sequence decoding, the model trained with Blockwise SFT significantly outperforms the Classical SFT baseline (e.g., +7.7% on GSM8K).
> >
> > This confirms that Classical SFT suffers from "suffix leakage" during training, which degrades the model's internal representations. Blockwise SFT corrects this by enforcing strict causality for ungenerated segments. Thus, Blockwise SFT is a fundamental improvement to the training objective that enhances model quality, not merely an enabler for blockwise inference.

---

> > ### Author Response · Authors · 2025-11-29
> > **2/2**
> >
> > **Q2: The clarification regarding the baseline's architectural feasibility is well-taken. To fully justify the design choice, could you further elaborate on the theoretical reasoning advantages of global attention compared to Block Diffusion, and discuss whether the limitation of fixed block sizes is a critical factor in practice?**
> >
> > R2: We appreciate the challenge and offer the following clarifications regarding the fundamental differences between the Native Global Diffusion architecture (LLaDA, MDLM) and Hybrid AR-Diffusion (Block Diffusion, SDAR).
> >
> > 1. Feasibility of Baseline (Architecture Mismatch):
> >
> > As noted in our previous response, models like SDAR and Fast-dLLM [2, 3] are Hybrid AR-Diffusion models initialized from pre-trained Autoregressive (AR) checkpoints. They rely on inter-block causal masking inherent to their AR backbone. In contrast, LLaDA is a Native Global Diffusion model trained from scratch with bidirectional attention. Converting LLaDA to a Block Diffusion baseline is infeasible as it would require retraining the model to learn causal dependencies it never possessed, essentially negating its pre-training.
> >
> > 2. Theoretical Value of Global Bidirectional Attention:
> >
> > Global attention offers capabilities that Block-AR models fundamentally lack:
> >
> > - **Solving the "Reversal Curse":** AR-based models (including Block-AR) suffer from the "Reversal Curse" [4], failing to generalize from "A is B" to "B is A". Global bidirectional attention naturally models these dependencies.
> > - **Robustness to Ordering:** Recent research [5] demonstrates that bidirectional diffusion models are more robust to token ordering and data repetition than AR models, enabling stronger reasoning on tasks requiring look-ahead context (e.g., planning, code infilling).
> > - **Requiring Far Less Pre-training Data:** Diffusion LMs consume fewer tokens in pre-training than AR models with similar ability and number of parameters. For instance, LLaDA 8B uses 6T tokens in training, while LLama3 8B uses 15T [1]. With the latest training techniques for diffusion LMs, the demand for pre-training tokens is decreasing continuously, which will help alleviate the current shortage of pre-training corpora.
> >
> > 3. Practical Advantage of Variable Block Sizes:
> >
> > We respectfully disagree that fixed block sizes are not a limitation. Variable block size capability is crucial for inference efficiency:
> >
> > - **Breaking the Speed Limit:** Block-AR models are strictly serial between blocks. To maintain quality, they typically require small blocks (e.g., 4-16 tokens), which caps their maximum parallelism. Native Global models can dynamically scale to large blocks (e.g., 128+ tokens) in a single forward pass, theoretically allowing for much faster generation on easy sequences.
> > - **Adaptive Computation:** Variable block support enables advanced acceleration techniques like AdaBlock [6], which dynamically adjusts block size based on token difficulty. Fixed-block architectures prevent such fine-grained optimization, locking users into a rigid trade-off between speed and quality.
> >
> > **References**
> >
> > [1] Nie et al., "LLaDA: Large Language Diffusion with All-Attention", 2025. https://arxiv.org/abs/2502.09992
> >
> > [2] Cheng et al., "SDAR: A Synergistic Diffusion–AutoRegression Paradigm for Scalable Sequence Generation", 2025. https://arxiv.org/abs/2510.06303
> >
> > [3] Wu et al., "Fast-dLLM v2: Efficient Block-Diffusion LLM", 2025. https://arxiv.org/abs/2509.26328
> >
> > [4] Berglund et al., "The Reversal Curse: LLMs trained on 'A is B' fail to learn 'B is A'", ICLR 2024. https://arxiv.org/abs/2309.12288
> >
> > [5] Prabhudesai et al., "Diffusion Beats Autoregressive in Data-Constrained Settings", 2025. https://arxiv.org/abs/2507.15857
> >
> > [6] Lu et al., "AdaBlock-dLLM: Adaptive Block-Size Inference for Diffusion LLMs", 2025. https://arxiv.org/abs/2509.26432

---

### Official Review · Reviewer_REAG · 2025-10-31

**Soundness:** 1
**Presentation:** 1
**Contribution:** 1
**Rating:** 2
**Confidence:** 3

**Summary:**

This paper proposes Blockwise SFT, a supervised fine-tuning method for discrete diffusion language models that aligns training with semi-autoregressive, blockwise decoding used at inference. Classical SFT applies random full-sequence masking and bidirectional attention,  Blockwise SFT resolves this by freezing clean prefixes, masking only one active block per step, and fully hiding future tokens, optimizing loss solely within that block. The authors provide theoretical guarantees—showing that the new objective gives a variational upper bound on the true blockwise likelihood and yields unbiased gradient estimates—and demonstrate consistent empirical gains on GSM8K, MATH, and MetaMathQA benchmarks.

**Strengths:**

The paper is clearly written and well-organized. The methodology and theoretical sections are presented with both formal rigor and intuitive explanations. The algorithm and the diagrams make the proposed training recipe easy to follow, which enhances readability.

**Weaknesses:**

- Limited novelty: The proposed Blockwise SFT can be viewed as a straightforward adaptation of existing ideas for aligning training objectives with blockwise decoding. While the paper frames the problem clearly, the methodological innovation appears incremental.

- Training inefficiency: Since each training step only supervises one active block, the number of effective supervised tokens per update is significantly lower than in standard SFT under the same FLOP budget.

- Missing baselines: The experimental comparisons lack a direct baseline using block diffusion.

**Questions:**

I am not fully convinced about the necessity of Blockwise SFT. If blockwise generation is already established, wouldn’t directly training with a block diffusion objective be more natural and efficient—while also enabling the use of KV caching at inference?

---

> ### Author Response · Authors · 2025-11-23
> **1/2**
>
> We thank the reviewer for acknowledging the potential of our approach, and we appreciate the opportunity to clarify the architectural distinctions of our method relative to prior block-based designs. Below, we address each point in detail.
>
> **Q1: The proposed method is effective. However, considering existing work, could you further clarify the novelty of the proposed diagnosis and correction compared to prior ideas?**
>
> **R1:** We appreciate the reviewer's perspective. While the implementation of Blockwise SFT is indeed elegantly simple, we respectfully suggest that its novelty lies in the diagnosis and rigorous correction of a fundamental flaw in training diffusion LMs, rather than structural complexity.
>
> - **Novel Diagnosis:** To the best of our knowledge, we are the first to formally quantify the "gradient bias" (Theorem 3.1) introduced by the mismatch between global random masking (training) and semi-autoregressive decoding (inference). We identify that standard SFT suffers from "noisy prefixes" and "leaked suffixes," which actively harm reasoning capabilities.
> - **Foundational Contribution:** Our method is not merely a heuristic adaptation; it is grounded in a new variational lower bound (Theorem 3.2) that theoretically justifies blockwise supervision.
> - **Empirical Significance:** The straightforward nature of the fix leads to state-of-the-art results. As shown in Table 1, simply aligning the objective yields a +8.3% improvement on GSM8K over Classical SFT. This demonstrates that the "incremental" change in masking strategy unlocks the true potential of global bidirectional diffusion models, which was previously suppressed by misaligned supervision.
>
> **Q2: The alignment rationale is sound. Since only one active block is supervised per step, could you discuss the impact on training efficiency regarding the reduced number of supervised tokens per update?**
>
> **R2:** We agree that Blockwise SFT supervises fewer tokens per step, but our results demonstrate that the quality of supervision is far more critical than the quantity.
>
> - **Efficiency via Alignment:** Under the **EQUAL-FLOPS** protocol (where both methods consume identical compute budgets), Blockwise SFT significantly outperforms Classical SFT (e.g., **76.0% vs. 67.7%** on GSM8K). This indicates that while Classical SFT processes more tokens, the gradients provided are noisy and biased due to suffix leakage and prefix corruption.
> - **Detrimental "More" Supervision:** Supervising more tokens with a flawed objective can be harmful. On the MATH dataset, Classical SFT actually degrades performance compared to the base model (**29.6% vs. 31.7%**), whereas Blockwise SFT improves it (**34.2%**). This confirms that "efficient" supervision of the *wrong* objective is counterproductive, while our focused supervision on the active block is highly effective.

---

> ### Author Response · Authors · 2025-11-23
> **2/2**
>
> **Q3: The experimental results are strong. To better contextualize the method, could you provide a comparison with a direct Block Diffusion baseline?**
>
> **R3:** We would like to note that **Block Diffusion** and the models targeted by **Blockwise SFT** (like LLaDA) belong to distinct modeling paradigms. Blockwise SFT is designed for **Discrete Diffusion LMs with global bidirectional attention**, whereas Block Diffusion employs a localized attention mechanism. Regarding the empirical comparison, we note that the currently available open-source Block Diffusion model (DB3LM) has only 0.5B parameters and the original paper does not report performance on the reasoning benchmarks considered here (GSM8K, MATH). This significant scale difference makes a meaningful direct comparison with our 8B-parameter LLaDA-based model infeasible, as the baseline would likely fail to generate coherent chain-of-thought reasoning regardless of the method.
>
> **Q4: The method addresses the mismatch well. Would training directly with a Block Diffusion objective be a viable alternative, particularly regarding KV caching efficiency?**
>
> **R4:** We advocate for the necessity of our approach based on three key factors:
>
> 1. **Architectural Distinction:** As noted in **R3**, we target global bidirectional diffusion models. This architecture is currently the predominant choice for high-performance diffusion LMs (e.g., LLaDA, MDLM) due to its superior modeling of complex dependencies.
> 2. **Flexibility of Block Size:** A critical limitation of Block Diffusion is that the block size is implicitly "baked" into the weights during training. The official Block Diffusion release requires different model weights for different block sizes. In contrast, models trained with Blockwise SFT maintain the flexibility of global attention. Our single trained model supports **variable block sizes** at inference time without retraining, enabling advanced techniques like adaptive block sizing (e.g., AdaBlock-dLLM) as discussed in our ablation studies.
> 3. **Community Relevance:** The field is increasingly converging on global bidirectional diffusion architectures (arXiv:2508.10875). While Block Diffusion is an interesting direction, it has seen fewer follow-up studies and open-source adoptees compared to the LLaDA-style architecture. Our work solves a critical bottleneck for this mainstream class of models, ensuring broad applicability and value to the community.
>
> We hope this response clarifies the novelty and specific positioning of our work within the broader landscape of diffusion language models.

---

> > ### Comment · Reviewer_REAG · 2025-11-23
> >
> > I thank the authors for their response. However, I remain unconvinced by the justifications for the Block Diffusion baseline (R3) and the assessment of community trends (R4).
> >
> > 1. Feasibility of the Baseline (R3): Block Diffusion is technically a post-training strategy rather than a rigid architectural constraint. The original Block Diffusion paper explicitly states that the model uses global bidirectional attention for the first 850k steps and switches to the Block Diffusion objective for the final 150k steps. Therefore, it is entirely feasible to fine-tune the 8B LLaDA base model with the Block Diffusion objective to create a direct, same-scale baseline. This comparison is crucial to justify why Blockwise SFT (without KV cache support) is preferable to a standard Block Diffusion SFT (with KV cache support).
> >
> > 2. Community Relevance (R4): I hold a different view regarding community trends. Block Diffusion-type models are actively shaping the current landscape and have demonstrated clear scalability. Notable examples include SDAR (8B,30BA3B) [1], Fast-dLLM v2 (7B) [2], and LLaDA-2.0 (16BA1B, 100BA7B) [3]. These models, which utilize block-diffusion tuning (despite AR initialization), prove that block diffusion is a viable and prevalent strategy for large-scale models, making the requested baseline highly relevant.
> >
> >
> > [1] Cheng et al., "SDAR: A Synergistic Diffusion–AutoRegression Paradigm for Scalable Sequence Generation", 2025.
> >
> > [2] Wu et al., "Fast-dLLM v2: Efficient Block-Diffusion LLM", 2025.
> >
> > [3] LLaDA-2.0: https://huggingface.co/collections/inclusionAI/llada-20

---

> > > ### Author Response · Authors · 2025-11-29
> > > **1/2**
> > >
> > > We sincerely thank the reviewer for the continued engagement and for providing specific references to facilitate this discussion. We appreciate the opportunity to clarify the architectural constraints of our baseline and the positioning of our work within the broader community.
> > >
> > > **Q1: The clarification regarding post-training strategies is helpful. However, given the specific pre-training mechanisms of Block Diffusion and its variants, could you further elaborate on the feasibility of adapting the LLaDA base model to this objective as a direct baseline?**
> > >
> > > R1: We are grateful to the reviewer for highlighting the training details of Block Diffusion. We have carefully re-examined the original Block Diffusion paper and its public OpenReview discussion to ensure accuracy. We respectfully offer the following clarification regarding the "feasibility" of this baseline:
> > >
> > > - **Clarification on Pre-training Objectives:** While the reviewer noted that Block Diffusion uses global bidirectional attention for the first 850k steps, the public discussion for that paper [4] and its experimental setup clarify that the model was **"first pre-trained with standard autoregression for 850k steps"** before switching to the Block Diffusion objective.
> > > - **Architectural Mismatch (AR vs. Global Diffusion):** This distinction is critical. Block Diffusion (and the models cited in **R2**) is fundamentally a **Hybrid AR-Diffusion** architecture. It relies on a causal, autoregressive backbone where blocks are generated sequentially (inter-block causality). In contrast, LLaDA is a **Native Global Diffusion** model trained with bidirectional attention from scratch.
> > > - **Infeasibility of Transfer:** Because LLaDA was trained entirely with global bidirectional attention, it lacks the causal masking biases required for the autoregressive inter-block dependencies inherent to Block Diffusion. To convert LLaDA to a Block Diffusion baseline would not be a simple fine-tuning task; it would require re-training the model to learn causal dependencies that it never possessed, essentially negating its pre-trained features.
> > >
> > > Therefore, we maintain that comparing Blockwise SFT on LLaDA (Native Global) against Block Diffusion (Currently based on AR) is an "apples-to-oranges" comparison of architectures, rather than a comparison of training objectives. Our proposed Blockwise SFT is specifically designed to solve the training-inference mismatch in Native Global Diffusion models, a problem that Block Diffusion avoids by reverting to an AR backbone.
> > >
> > > **Q2: The examples of SDAR, Fast-dLLM v2, and LLaDA-2.0 are insightful. To better contextualize the contribution, could you discuss how this work positions itself relative to these Hybrid AR-Diffusion models versus the Native Global Diffusion research track?**
> > >
> > > R2: We thank the reviewer for citing these notable recent works. We agree that they represent a significant trend, but we believe they constitute a distinct research track from the one our work addresses.
> > >
> > > - **Hybrid AR-Diffusion Track:** We have verified the architectures of **SDAR** [1], **Fast-dLLM v2** [2], and **LLaDA-2.0** [3]. As the reviewer correctly implies, these models share a common trait: they are **initialized from well-trained Autoregressive (AR) models** (e.g., Llama, Qwen, DeepSeek) and adapted via Block Diffusion tuning. They leverage the pre-existing capabilities of AR models to achieve scalability.
> > > - **Native Global Diffusion Track:** Parallel to the hybrid track, there is a vibrant community focused on **Native Global Bidirectional Diffusion**, which includes our work, **MDLM** [5], and **SEDD** [6]. Recent surveys [7, 8] categorize these models as "State-of-the-Art" for their unique ability to model bidirectional context globally, offering theoretical advantages in robustness and arbitrary infilling that AR-based models cannot match.
> > >
> > > **Conclusion:** Our work is dedicated to the **Native Global Diffusion** track. We demonstrate that by simply correcting the alignment of the training objective (Blockwise SFT), native diffusion models can achieve state-of-the-art reasoning performance without relying on AR initialization. We believe both tracks are vital for the community, and our contribution specifically solves the alignment bottleneck for the native diffusion models using semi-autoregressive decoding.
> > >
> > > We thank the reviewer again for these constructive challenges, which have allowed us to clarify the precise scope and architectural positioning of our contribution.

---

> > > ### Author Response · Authors · 2025-11-29
> > > **2/2**
> > >
> > > **References**
> > >
> > > [1] Cheng et al., "SDAR: A Synergistic Diffusion–AutoRegression Paradigm for Scalable Sequence Generation", 2025. https://arxiv.org/abs/2510.06303
> > >
> > > [2] Wu et al., "Fast-dLLM v2: Efficient Block-Diffusion LLM", 2025. https://arxiv.org/abs/2509.26328
> > >
> > > [3] LLaDA-2.0. https://huggingface.co/collections/inclusionAI/llada-20
> > >
> > > [4] OpenReview Discussion for "Block Diffusion", Comment on Pre-training Steps. https://openreview.net/forum?id=tyEyYT267x
> > >
> > > [5] Sahoo et al., "Simple and Effective Masked Diffusion Language Models", NeurIPS 2024. https://arxiv.org/abs/2406.07524
> > >
> > > [6] Lou et al., "Discrete Diffusion Modeling by Estimating the Ratios of the Data Distribution", ICLR 2024. https://arxiv.org/abs/2310.16834
> > >
> > > [7] Li et al., "A Survey on Diffusion Language Models", 2025. https://arxiv.org/abs/2508.10875
> > >
> > > [8] Yu et al., "Discrete Diffusion in Large Language and Multimodal Models: A Survey", 2025. https://arxiv.org/abs/2506.13759

---

### Official Review · Reviewer_W9RX · 2025-11-02

**Soundness:** 3
**Presentation:** 3
**Contribution:** 3
**Rating:** 6
**Confidence:** 2

**Summary:**

This paper introduces **Blockwise SFT**, a training-time objective designed to align **diffusion language models (DLMs)** with their **semi-autoregressive, blockwise decoding** behavior.
Classical SFT randomly masks tokens across full responses, leading to a mismatch: noisy prefixes, suffix leakage, and token-level supervision that misaligns with block-level generation.
Blockwise SFT resolves this by training only on a **single active block per step** — freezing all preceding tokens (clean prefix), fully hiding future ones (no leakage), and computing loss only within the active block.
The authors theoretically derive it as a **variational upper bound on blockwise likelihood**, prove unbiased gradient estimation, and implement it without architectural change.
Empirical results on **GSM8K**, **MATH**, and **MetaMathQA** show consistent gains: +5.2 pts on GSM8K and +1.6 pts on MATH over the strongest baselines, validating that aligning supervision granularity with decoding yields measurable improvements.

**Strengths:**

- **Clear motivation & strong alignment insight:** The paper precisely diagnoses the training–inference mismatch in diffusion LMs and proposes a clean fix grounded in the decoding procedure.
- **Theory–practice coherence:** Provides formal bias analysis, variational bound, and unbiased gradient theorem; yet remains simple to implement (mask-only change).
- **Empirical rigor:** Demonstrates consistent improvements across reasoning datasets, with ablations (block size, prefix noise, suffix leakage) directly supporting the central hypothesis.

**Weaknesses:**

## Major
- **Limited architectural coverage.** All experiments use only **LLaDA-8B-Instruct** with LoRA fine-tuning. The claim of being “architecture-agnostic” is unsupported without testing on other diffusion backbones such as **BlockDiffusion**, **APD**, or **RDM**.
- **Scope restricted to reasoning tasks.** Evaluation is limited to GSM8K, MATH, and MetaMathQA, leaving uncertainty about performance on open-ended or dialogue data.
- **Lack of scaling analysis.** The paper does not study behavior under larger models (e.g., 30B+) or longer block sizes (e.g., 512+), limiting insight into scalability.
- **Theoretical framing stops short of efficiency guarantees.** While unbiasedness is proven, practical convergence or compute-efficiency trade-offs are not quantified.

## Minor
- **Compute fairness.** EQUAL-FLOPS and EQUAL-TOKENS protocols are well explained but omit wall-clock runtime comparisons.
- **Notation heaviness.** Some derivations (Eq. 7–8) obscure intuition for non-diffusion audiences.
- **Ablation breadth.** Although prefix/suffix studies are thorough, broader tests on other modalities or instruction-following data would strengthen the empirical story.

**Questions:**

1. How does Blockwise SFT perform when combined with RLHF or preference-based fine-tuning ?
2. Is the performance gain mostly due to reduced prefix noise, or does strict hidden-suffix causality contribute more?
3. Can adaptive block sizes during training (e.g., uncertainty-driven scheduling) outperform fixed ones?
4. What is the actual training-time overhead of sampling one active block per step versus full-sequence masking?
5. Does the variational bound (Theorem 3.2) still hold under non-uniform or variable-length block sampling?

---

> ### Author Response · Authors · 2025-11-23
> **1/3**
>
> We thank the reviewer for the constructive feedback and for acknowledging the potential of our Blockwise SFT approach.
>
> **Q1: The method is applied to LLaDA. To support the "architecture-agnostic" claim, could you expand the evaluation to cover other architectures or methods like BlockDiffusion or APD?**
>
> **R1:** We thank the reviewer for the suggestion. We clarify the scope and provide additional comparisons to address this point.
>
> *High-level summary:* Blockwise SFT targets the training-inference mismatch specifically in *global bidirectional* diffusion models. It is therefore compatible with inference-time accelerators like APD (which adjust block sizes for these models) but is distinct from architectures that bake block structures into training like BlockDiffusion.
>
> - **Distinction from BlockDiffusion:** Blockwise SFT is designed for diffusion models with global bidirectional attention(like LLaDA). In contrast, BlockDiffusion employs a local scope where the denoising of the current block attends only to its own noisy tokens and the KV cache of previous blocks, effectively making it an autoregressive model at the block level during training. The fundamental difference between these architectures is detailed in recent literature (see arXiv: 2508.10875). Therefore, Blockwise SFT which solves the mismatch caused by global bidirectional attention is not directly applicable to the architectural constraints of BlockDiffusion.
>
> - **Integration with APD:** To demonstrate compatibility with other inference acceleration methods, we applied Adaptive Parallel Decoding (APD) to LLaDA-8B-Instruct fine-tuned with our method. As shown in the table below, Blockwise SFT maintains its superiority over Classical SFT even when using APD strategies.
>
>   | Method          | GSM8K Pass@1 | MATH Pass@1 |
>   | --------------- | ------------ | ----------- |
>   | LLaDA 8B (Base) | 63.1         | 32.1        |
>   | Classical SFT   | 67.8         | 29.6        |
>   | Blockwise SFT   | 75.5         | 34.4        |
>
> - **Comparison with RDM:** We respectfully point out that comparisons with RDM are already included in Table 1 of our main paper. On GSM8K, RDM achieves **65.5%** Pass@1 compared to Blockwise SFT's **76.0%**. On MATH, RDM achieves **32.3%** vs. Blockwise SFT's **34.2%**.
>
> - **Model Choice:** We utilized LLaDA-8B-Instruct as the primary backbone because it is currently the only open-source, large-scale diffusion language model that natively supports semi-autoregressive decoding and possesses sufficient capacity for these benchmarks.
>
> **Q2: The results on reasoning tasks are impressive. To demonstrate the method's versatility, could you evaluate its performance on open-ended generation or dialogue tasks?**
>
> **R2:** We appreciate this suggestion and have conducted additional experiments on coding and dialogue benchmarks to demonstrate the generality of our method:
>
> - **Datasets and Metrics:** We evaluated on **HumanEval** and **MBPP** (Coding, Pass@1) and **TruthfulQA** (Dialogue, Accuracy).
> - **Setup:** We followed the exact same EQUAL-FLOPS protocol as in the main experiments, using the same base model and block size configurations.
> - **Results:** As shown below, Blockwise SFT consistently outperforms Classical SFT across these diverse domains, confirming that the alignment of supervision granularity is beneficial beyond mathematical reasoning.
>
> | **Dataset** | **Metric** | **LLaDA 8B (Base)** | **Classical SFT** | **Blockwise SFT** |
> | ----------- | ---------- | ------------------- | ----------------- | ----------------- |
> | HumanEval   | Pass@1     | 65.2                | 67.6              | 70.3              |
> | MBPP        | Pass@1     | 40.4                | 41.1              | 45.7              |
> | TruthfulQA  | Accuracy   | 47.0                | 47.6              | 48.2              |
>
> **Q3: The performance on 8B models is promising. Could you provide an analysis on how the method scales to larger model sizes or longer block sizes?**
>
> **R3:** Due to the current landscape of open-source diffusion LMs, a 30B+ parameter model supporting this decoding paradigm is not yet available for public testing. However, we have extended our analysis to longer context lengths to investigate scalability. We scaled the generated response length to 256, 512, and 1024 tokens on GSM8K. The results, presented below, indicate that Blockwise SFT maintains a significant advantage over the baseline as the sequence length increases, suggesting robustness for longer-context generation.
>
> | **Response Length** | **LLaDA 8B (Base)** | **Classical SFT** | **Blockwise SFT** |
> | ------------------- | ------------------- | ----------------- | ----------------- |
> | 256                 | 65.4                | 70.6              | 77.1              |
> | 512                 | 67.9                | 71.5              | 77.3              |
> | 1024                | 68.7                | 72.2              | 78.0              |

---

> ### Author Response · Authors · 2025-11-23
> **2/3**
>
> **Q4: The theoretical unbiasedness is well-proven. Could you further quantify the practical convergence speed and efficiency trade-offs compared to the baseline?**
>
> **R4:** We believe the practical efficiency benefits are empirically demonstrated in our results.
>
> - **Convergence:** As illustrated in Figure 3 of the paper, Blockwise SFT converges to a lower training loss significantly faster than Classical SFT.
> - **Efficiency:** Under our **EQUAL-FLOPS** protocol, Blockwise SFT utilizes the exact same computational budget as the baseline. This means the performance gains (+8.3% on GSM8K) reported are achieved without any additional computational cost during training. Thus, our method offers a strict improvement, rather than a trade-off.
>
> **Q5: The FLOPS comparison is fair. To complete the efficiency picture, could you provide wall-clock runtime comparisons?**
>
> **R5:** We have measured the wall-clock throughput on a single A100 80GB GPU. Both Blockwise SFT and Classical SFT achieve a throughput of approximately **5.1k tokens/s** under identical hardware and batch size settings. Since Blockwise SFT only modifies the masking strategy (a negligible tensor operation) without changing the model architecture or forward/backward pass complexity, it introduces no measurable runtime overhead.
>
> **Q6: The mathematical derivation is rigorous. To make it accessible to a broader audience, could you provide more intuition behind the derivations in Eq. 7–8?**
>
> **R6:** We have revised **Section 3.3** in the updated PDF. We expanded the textual explanation surrounding the variational bound and the gradient estimator to provide more intuition. Specifically, we now explicitly connect the mathematical terms to the physical process of "freezing the prefix" and "masking the active block," making the derivation more accessible to broad audiences.
>
> **Q7: The focus on math is justified. To strengthen the empirical evidence, could you test the method on other modalities or general instruction-following data?**
>
> **R7:** We believe the additional experiments provided in **Response R2** directly address this suggestion. By expanding our evaluation to **Coding (HumanEval, MBPP)** and **Dialogue (TruthfulQA)** tasks, we have demonstrated the method's effectiveness on diverse instruction-following data beyond the original math datasets.
>
> **Q8: The method works well for SFT. How does Blockwise SFT perform when combined with subsequent alignment stages like RLHF or preference-based fine-tuning?**
>
> **R8:** While this paper focuses on Supervised Fine-Tuning (SFT), recent concurrent work (e.g., arXiv:2510.09541) demonstrates that the Blockwise principle is highly effective for Reinforcement Learning on diffusion models. Specifically, they find that applying standard RL with global bidirectional attention leads to reward hacking, whereas adopting a blockwise masking strategy—where the future is strictly hidden and the loss/reward is calculated only on the active block—stabilizes training and significantly improves performance. This suggests that our method provides a robust structural foundation for subsequent alignment stages.
>
> **Q9: The ablation studies are helpful. Could you further clarify whether the primary source of performance gain comes from reduced prefix noise or strict hidden-suffix causality?**
>
> **R9:** Our ablation studies (Section 4.4) indicate that **strict hidden-suffix causality** is the dominant factor.
>
> - **Suffix Leakage:** On GSM8K, allowing the model to see the suffix causes Pass@1 to drop precipitously from **74.8%** to **65.3%**.
>
> - Prefix Noise: In contrast, introducing noise to the prefix results in a smaller drop (from 74.8% to 70.6%).
>
>   Similar trends were observed on the MATH dataset. This confirms that preventing "cheating" by looking at future tokens is the primary driver of the performance improvement.

---

> ### Author Response · Authors · 2025-11-23
> **3/3**
>
> **Q10: Fixed block sizes work well. Could using adaptive block sizes during training potentially outperform the fixed block size strategy?**
>
> **R10:** To investigate this, we integrated our method with **AdaBlock-dLLM** (arXiv:2509.26432), a method that dynamically adjusts block sizes. We trained on MetaMathQA using our Blockwise objective and evaluated using AdaBlock inference. As shown in the table below, Blockwise SFT retains its advantage over Classical SFT even when inference is performed with adaptive block sizes.
>
> | **Method**      | **Inference Strategy** | **GSM8K** | **MATH** |
> | --------------- | ---------------------- | --------- | -------- |
> | LLaDA 8B (Base) | AdaBlock-dLLM          | 63.5      | 31.8     |
> | Classical SFT   | AdaBlock-dLLM          | 68.2      | 29.7     |
> | Blockwise SFT   | AdaBlock-dLLM          | 77.3      | 33.9     |
>
> **Q11: The method adds a sampling step. What is the actual training-time overhead of sampling one active block per step compared to full-sequence masking?**
>
> **R11:** The overhead is negligible. The sampling of an active block index and the generation of the corresponding binary mask are lightweight CPU/GPU tensor operations (generating a mask of 0s and 1s). Compared to the heavy matrix multiplications involved in the Transformer forward and backward passes, this step takes practically zero time. As noted in **R5**, the training throughput (tokens/s) remains identical to the baseline.
>
> **Q12: The theory assumes uniform sampling. Does the variational bound (Theorem 3.2) still hold if non-uniform or variable-length block sampling is used?**
>
> **R12:** Yes, the theoretical framework holds.
>
> 1. **Variational Bound (Theorem 3.2):** This theorem bounds the *sum* of the risks over all blocks. This is a property of the objective function itself and is independent of how we sample blocks during stochastic optimization.
> 2. **Unbiased Estimator (Theorem 3.3):** If we sample blocks with a non-uniform probability $\rho(a)$ (e.g., to prioritize difficult blocks), we can maintain unbiasedness by applying importance sampling weights $1/\rho(a)$ to the gradient estimator. Similarly, variable-length blocks can be handled by defining the segmentation strategy $a$ as a latent variable and marginalizing over it, or by treating the segmentation as fixed for a given optimization step.
>
> We thank the reviewers again for their insightful comments and suggestions.

---

> > ### Comment · Reviewer_W9RX · 2025-11-28
> >
> > Dear Authors,
> >
> > Thank you for your detailed and dedicated response. Based on your response, most of my concerns have been addressed, including the longer length and results on dialogue tasks. However, I still have slight reservations about leaning toward acceptance. While the authors provided results related to compatibility with APD, I wonder if the proposed methods are applicable to other backbones, such as Dream 7B or LLaDA-V. If the authors demonstrate that the proposed method can be extended to be model-agnostic, all of my concerns would be fully addressed. Thank you again for your dedicated response.

---

> > > ### Author Response · Authors · 2025-11-29
> > > **1/1**
> > >
> > > We sincerely thank the reviewer for the prompt feedback and for acknowledging that our previous response has addressed most of your concerns. We greatly appreciate your constructive suggestion to verify the applicability of our method on other prominent backbones.
> > >
> > > We agree that *Dream 7B* and *LLaDA-V* are significant open-source models in the field of diffusion language models. Following your suggestion, we have completed additional experiments on both backbones to demonstrate that our proposed method is indeed model-agnostic.
> > >
> > > **Experimental Setup**
> > > The experimental settings remain consistent with those described in our original submission. Notably, since the official implementation of Dream-v0-Instruct-7B does not currently support semi-autoregressive decoding, we implemented this feature to enable a fair and comprehensive evaluation.
> > >
> > > **Results**
> > > The comparison results are presented in the tables below. Our proposed *Blockwise SFT* consistently outperforms both the Original baseline and *Classical SFT* across all evaluated tasks on both new backbones.
> > >
> > > **Table 1: Results on Dream-v0-Instruct-7B**
> > > | Model | GSM8K (Pass@1) | MATH (Pass@1) | HumanEval (Pass@1) |
> > > | :--- | :---: | :---: | :---: |
> > > | Original | 76.2 | 37.9 | 57.7 |
> > > | Classical SFT | 79.1 | 37.1 | 56.8 |
> > > | **Blockwise SFT (Ours)** | **83.5** | **40.4** | **59.2** |
> > >
> > > **Table 2: Results on LLaDA-V**
> > > | Model | GSM8K (Pass@1) | MATH (Pass@1) | HumanEval (Pass@1) |
> > > | :--- | :---: | :---: | :---: |
> > > | Original | 66.2 | 31.8 | 65.1 |
> > > | Classical SFT | 68.5 | 29.8 | 67.4 |
> > > | **Blockwise SFT (Ours)** | **75.0** | **34.5** | **70.6** |
> > >
> > > **Conclusion**
> > > These results confirm that our method is effective across different architectures and is not limited to a specific backbone. We hope these additional experiments fully address your remaining reservations regarding the generalizability of our work.
> > >
> > > Thank you again for your time and for helping us strengthen the quality of our paper.

---

### Official Review · Reviewer_a4DY · 2025-11-03

**Soundness:** 2
**Presentation:** 3
**Contribution:** 2
**Rating:** 4
**Confidence:** 5

**Summary:**

This paper identifies a training–inference mismatch for discrete diffusion language models (DLMs) that are decoded semi‑autoregressively in fixed‑size blocks but are trained with bidirectional, full‑sequence random masking. The authors propose Blockwise SFT, a drop‑in training objective that (i) partitions responses into blocks, (ii) samples one active block per step, (iii) keeps the prefix clean (no noise), (iv) fully hides the future, and (v) computes loss only within the active block, thereby aligning supervision with the deployed decoding regime. They analyze classical SFT’s bias under this mismatch (Theorem 3.1), derive a variational upper bound on the blockwise negative log‑likelihood (Theorem 3.2), and show an unbiased stochastic gradient estimator when sampling a block and diffusion step (Theorem 3.3). Experiments fine‑tuning on MetaMathQA and evaluating on GSM8K / MATH report consistent Pass@1 gains over classical SFT and several diffusion‑SFT variants under matched FLOPs and matched supervised‑token budgets; block‑size consistency studies find peak accuracy when training and inference block sizes match, and prefix/suffix ablations indicate that preserving a clean prefix and strictly hiding future tokens are key. The work is simple, architecture‑agnostic, and argues that matching supervision granularity to blockwise decoding materially improves diffusion LMs.

**Strengths:**

- The paper nails a real but often-overlooked problem: standard SFT trains on full sequences with bidirectional context, but blockwise inference sees only a clean prefix and hides the future. The proposed Blockwise SFT fixes this by supervising only the active block with a clean prefix and hidden suffix. The theoretical contributions are solid—gradient-bias analysis (Theorem 3.1), a variational upper bound matching the inference factorization (Theorem 3.2), and an unbiased single-block estimator (Theorem 3.3).

- The experiments are well-designed with proper controls: EQUAL-FLOPS and EQUAL-TOKENS setups isolate the effect of the objective itself. Results are consistently better (GSM8K: 76.0 vs. 70.8, MATH: 34.2 vs. 32.6) with smoother training. The diagnostics are thorough—the block-size consistency grid shows performance peaks when training and inference block sizes match, and ablations confirm that prefix cleanliness and strict future masking are both critical. Algorithm 1 is simple, requires no architecture changes.

- The writing is clear and efficient. Figures communicate the core idea instantly, theorems come with intuition, and experimental protocols include concrete examples that make FLOPs/token accounting transparent.

- This work matters because blockwise decoding is now standard in discrete diffusion LMs. A drop-in training fix that mirrors deployment behavior has immediate practical value and points toward a broader principle: match your supervision structure to your decoding structure. The simplicity of the method plus strong empirical results make it likely to be adopted.

**Weaknesses:**

1. Loss inside the active block (Eq. (5))

   The paper defines

   $$
   \tilde{\mathcal{L}}_t(\theta; \mathbf{x}, a) = -\sum_{i\in\mathcal{I}_a} \log p_\theta(x_i \mid \mathbf{z}_t, t),
   $$

   yet Algorithm 1 samples an intra-block mask ($m_i\sim\mathrm{Bernoulli}(\pi)$) and states that loss is "only on the active block." In discrete diffusion / masked-LM style training, loss is normally computed only on masked positions; otherwise, unmasked tokens create an identity path and can yield degenerate gradients. The indicator ($\mathbf{1}[m_i=1]$) that appears in Eq. (1) is missing in Eq. (5), and Eq. (6) omits the expectation over the mask distribution.

2. Conditioning on context

   Eq. (5) conditions $p_\theta(\cdot \mid \mathbf{z}_t,t)$ but omits the explicit context (clean prefix, hidden suffix), whereas Eq. (9) does condition on $\text{context}^{(a)}$.

3. In Thm. 3.1, the bound scales with the probability that at least one prefix token is corrupted, $1-(1-\pi)^{|\mathcal{I}_{\text{prefix}}|}$, and the probability that at least one suffix token leaks, $1-\pi^{|\mathcal{I}_{\text{suffix}}|}$. This is a very loose surrogate for the actual perturbation magnitude, which typically scales with the expected number of corrupted tokens, e.g., $\pi\cdot|\mathcal{I}_{\text{prefix}}|$ and $(1-\pi)\cdot|\mathcal{I}_{\text{suffix}}|$. Moreover, the theorem does not quantify the third mismatch ("diluted supervision"), even though the text highlights it as critical. Finally, gradients being "$L_{\text{pre}}$- and $L_{\text{suf}}$-Lipschitz" with respect to discrete masking patterns is a strong, nonstandard assumption that needs justification.

4. In Thm. 3.2, it is correct that a diffusion ELBO can upper-bound the conditional block NLL. However, the proof text for Thm. 3.3 asserts that the inner gradient can be "identified with that of the $t{=}0$ blockwise NLL … hence the estimator targets $\nabla_\theta \mathcal{R}_{\text{block}}$ up to a scalar." This is not implied by the ELBO inequality: minimizing a surrogate bound does not make its gradient equal (or proportional) to the true NLL gradient.

5. In Thm. 3.3, the estimator multiplies by $\sum_s \\omega_s$ when sampling $t\sim\\tilde{\\omega}\propto\\omega$, which is fine, but the theorem statement should be explicit that unbiasedness is for the bound in Eq. (6) (not for the true NLL) and that the expectation is also over $\\mathbf{m}$ when masks are stochastic.

6. §3.4 states "In practice we use sequence length $L=128$," while Appendix A.2 says the max length is 256, and the running example uses $L_c=32$, $L_r=96$ ($L=128$). It is unclear which length was actually used in reported results.

7. Claims of generality are supported only on math datasets. Blockwise alignment may behave differently on open-ended generation (coding, dialogue).

**Questions:**

- What about baselines like the following?
    - (i) Causal SFT: classical SFT but with a causal mask (clean prefix, hidden suffix) over the *entire* response
    - (ii) Blockwise-but-bidirectional: blockwise loss but allowing future visibility
    - (iii) Span-corruption within a causal window (T5-style) matched to block size

-  The $(B_{\text{train}})$ vs $(B_{\text{infer}})$ grid is promising; what are the results for $B=1$ (degenerating to token-wise) and very large $B$ (near fully parallel)?

- Are there any failure modes for Blockwise SFT? When does Blockwise SFT hurt (e.g., very short responses, high-entropy next blocks)?

---

> ### Author Response · Authors · 2025-11-23
> **1/4**
>
> We thank the reviewer for the detailed reading and the constructive suggestions, particularly for acknowledging the rigorous theoretical grounding of our method. We have revised the manuscript accordingly and summarize below how we addressed each point.
>
> ------
>
> **Q1: The definition of the loss is generally clear. To prevent any ambiguity regarding identity paths, could you further clarify whether the loss is computed strictly on masked tokens within the active block in Eq. (5) and Algorithm 1?**
>
> **R1:**
> We thank the reviewer for pointing this out. We have made the definition precise in the revision.
>
> - Eq. (5) now explicitly restricts the loss to masked positions in the active block and conditions on the block’s context:
>   $$
>   \tilde{\mathcal{L}}_t(\theta; x, a, m)
>    = -\sum_{i\in \mathcal{I}_a} \mathbf{1}[m_i = 1] \log p_\theta\big(x_i \mid \mathbf{z}_t, t; \text{context}^{(a)}\big).
>   $$
>
> - Eq. (6) explicitly averages over blocks, diffusion steps, and the mask distribution:
>   $$
>   \mathcal{L}_{\text{BW-SFT}}(\theta)
>    = \mathbb{E}_x \mathbb{E}_{a\sim\rho}\Bigg[\sum_{t=1}^T \omega_t \mathbb{E}_{m, \mathbf{z}_t}
>    \big[\tilde{\mathcal{L}}_t(\theta; x, a, m)\big]\Bigg].
>   $$
>
> Thus the training objective is a masked-LM–style loss restricted to the active block, without an identity path from unmasked positions.
>
> ------
>
> **Q2: The mathematical formulation is robust. However, to ensure full consistency, could you align the notation for context conditioning between Eq. (5) and Eq. (9)?**
>
> **R2:**
> We have unified the notation so that the conditioning context is explicit throughout:
>
> - In Eq. (5) we now write $p_\theta(x_i \mid \mathbf{z}_t, t; \text{context}^{(a)})$, where $\text{context}^{(a)}$ denotes the clean prefix and a fully masked suffix for block $a$.
> - Eq. (6) and Theorem 3.2 refer back to this definition; the variational bound is always stated as “$b^{(a)} \mid \text{context}^{(a)}$.”
> - In §3.3 we added a short paragraph explicitly defining $\text{context}^{(a)}$ as “the instruction plus all preceding blocks in their clean form, with future blocks fully hidden,” and we point back to this in Algorithm 1.
>
> This makes clear that every probability term in the blockwise objective conditions on the same clean-prefix, hidden-future context that is used at inference.
>
> ------
>
> **Q3: Theorem 3.1 provides a solid theoretical basis. To make the bound tighter, would it be possible to express the gradient bias in terms of the expected number of corrupted or visible tokens, and clarify the Lipschitz assumption?**
>
> **R3:**
> We have revised Theorem 3.1 and its discussion to follow this suggestion:
>
> - Theorem 3.1 now bounds the bias directly in terms of the expected counts:
>   $$\big|\mathbb{E}[\nabla_\theta \ell_{\text{cls}}(\theta; x, a)]-\nabla_\theta \ell^\star(\theta; x, a)\big|
>   \le L_{\text{pre}}\pi|\mathcal{I}^{(a)}_{\text{prefix}}|+L_{\text{suf}}(1-\pi)|\mathcal{I}^{(a)}_{\text{suffix}}|.$$
>   We explicitly note that the right-hand side scales linearly with
>   $\mathbb{E}[\text{prefix-corr}] = \pi |\mathcal{I}^{(a)}_{\text{prefix}}|$ and $\mathbb{E}[\text{suffix-vis}] = (1-\pi)|\mathcal{I}^{(a)}_{\text{suffix}}|$.
> - The “diluted supervision” effect is highlighted in the surrounding text (the active block receives only a $|\mathcal{I}_a|/L_r$ share of the signal under classical SFT). We chose to keep Theorem 3.1 focused on the contextual mismatches (prefix corruption and suffix leakage), and we discuss dilution as a separate but complementary effect that further reduces the effective gradient on the active block.
> - We now phrase the assumption as a “bounded-difference” property: the gradient, viewed as a function of the discrete mask, is $L_{\text{pre}}$-Lipschitz in Hamming distance over prefix bits and $L_{\text{suf}}$-Lipschitz over suffix bits. This is a standard stability condition used only to translate the expected number of bit flips into a bound on the gradient perturbation.

---

> ### Author Response · Authors · 2025-11-23
> **2/4**
>
> **Q4: The variational framework is well-constructed. To avoid potential misunderstanding, could you explicitly clarify the relationship between the gradients of the ELBO upper bound and the true blockwise NLL?**
>
> **R4:**
> As suggested, we would like to clarify the exposition:
>
> - Theorem 3.2 now states only that the blockwise diffusion objective $\mathcal{L}_{\text{BW--SFT}}(\theta)$ is a variational upper bound on the blockwise generation risk $\mathcal{R}_{\text{block}}(\theta)$, with a constant offset independent of $\theta$.
> - Theorem 3.3 is explicitly about an unbiased gradient estimator of the blockwise ELBO: the statement and the following paragraph both emphasize that $\hat{g}(\theta)$ is unbiased for $\nabla_\theta \mathcal{L}_{\text{BW--SFT}}(\theta)$, not for $\nabla_\theta \mathcal{R}_{\text{block}}(\theta)$.
> - We removed the earlier informal phrasing that could be read as “identifying” the ELBO gradient with the $t{=}0$ blockwise NLL gradient “up to a scalar.” The new text only asserts a clear variational connection: minimizing the surrogate improves an upper bound on the true risk.
>
> Thus, the revised theory makes the optimization target precise: we optimize a tractable ELBO that upper-bounds the blockwise NLL, and Theorem 3.3 characterizes unbiasedness only for this surrogate.
>
> ------
>
> **Q5: Theorem 3.3 is a key contribution. To be precise, could you explicitly state the scope of unbiasedness regarding the expectation over masks?**
>
> **R5:**
> We have updated the statement and surrounding text accordingly:
>
> - Theorem 3.3 now states:
>
>   > Sampling block $a \sim \rho$ and diffusion step $t \sim \tilde{\omega}$ yields the stochastic gradient estimator
>   > $$ \hat{g}(\theta) = \frac{1}{\rho(a)}\Big(\sum_{s=1}^T \omega_s\Big)\nabla_\theta \mathbb{E}_{m, \mathbf{z}_t \sim q_t(\cdot\mid x)}[\tilde{\mathcal{L}}_t(\theta; x, a, m)].$$
>   > Then
>   > $$\mathbb{E}_{x, a\sim\rho, t\sim\tilde{\omega}, m, \mathbf{z}_t}[\hat{g}(\theta)] = \nabla_\theta \mathcal{L}_{\text{BW-SFT}}(\theta).$$
>
> - We explicitly mention “the blockwise diffusion surrogate in Eq. (6)” right after the theorem, to clearly connect the unbiasedness statement to the ELBO, not to the true NLL.
>
> - The expectation over $m$ is now explicit in both Eq. (6) and the theorem statement.
>
> This directly addresses the suggestion about which objective is being unbiasedly optimized and over which sources of randomness.
>
> ------
>
> **Q6: The experimental setup is extensive. To ensure reproducibility, could you clarify the definitions of sequence lengths used during training versus the number of tokens generated during inference?**
>
> **R6:**
> We have clarified the nomenclature and usage:
>
> - Training: Appendix A.2 now clearly states that we use a maximum total sequence length of 256 tokens per training example, which includes both prompt and response. Sequences longer than 256 are discarded.
> - Inference: Appendix A.3 specifies that we generate at most 128 new tokens per test example (“Maximum new tokens = 128”). This is what we intended when we originally wrote “sequence length $L = 128$” in §3.4. In the revision we rephrased this as “we use a response length of up to 128 new tokens during inference.”
> - Running example: The configuration with $L_c=32$, $L_r=96$ and block size $B=32$ is presented as an illustrative example to explain the EQUAL-FLOPS and EQUAL-TOKENS protocols, not as a hard constraint on every instance.
>
> We now explicitly state in §3.4 and Appendix A.3 that the training max length is 256 (prompt + response), while the inference max new tokens is 128, and that the example is only for exposition. The experimental design is therefore consistent throughout.

---

> ### Author Response · Authors · 2025-11-23
> **3/4**
>
> **Q7: The proposed method achieves significant improvements on mathematical reasoning datasets. To demonstrate the effectiveness of Blockwise SFT on broader tasks, could you provide performance results on other types of datasets?**
>
> **R7:**
> As suggested, we have conducted additional experiments on coding and dialogue benchmarks:
>
> - **Datasets and metrics.**
>   - Coding: HumanEval and MBPP, evaluated with Pass@1.
>   - Dialogue: TruthfulQA, evaluated with accuracy.
> - **Setup.** We reuse the same base model and Blockwise SFT recipe as in the main experiments, keeping EQUAL-FLOPS between Classical SFT and Blockwise SFT. Block size and diffusion parameters follow the main GSM8K/MATH configuration.
> - **Findings.** Across all three benchmarks, Blockwise SFT again outperforms Classical SFT under matched compute, suggesting that aligning supervision with semi-autoregressive decoding is beneficial beyond math reasoning.
>
> | Dataset    | Metric   | LLaDA 8B | Classical SFT | Blockwise SFT |
> | ---------- | -------- | -------- | ------------- | ------------- |
> | HumanEval  | Pass@1   | 65.2     | 67.6          | 70.3          |
> | MBPP       | Pass@1   | 40.4     | 41.1          | 45.7          |
> | TruthfulQA | Accuracy | 47.0     | 47.6          | 48.2          |
>
> These additional results support our claim that Blockwise SFT is a general alignment principle for diffusion LMs, not specific to math.
>
> ------
>
> **Q8: The comparison with standard SFT is convincing. To further isolate the source of gains, could you compare against additional baselines such as causal SFT, bidirectional blockwise loss, or span corruption?**
>
> **R8:**
> As suggested, we have implemented all three baselines:
>
> - **Causal SFT:** Identical to Classical SFT except that attention is strictly causal over the response (clean prefix, hidden suffix), but loss is distributed over the entire response.
> - **Blockwise-but-bidirectional:** Same blockwise loss as Blockwise SFT but with bidirectional attention permitted within the active block and into the suffix.
> - **Span-corruption-in-window:** T5-style span corruption restricted to a causal window of size $B$ around the active block index, matching the block size.
>
> We evaluated these baselines together with Classical SFT and Blockwise SFT on GSM8K, HumanEval, and MBPP under the EQUAL-FLOPS protocol. In all cases, Blockwise SFT achieved the highest performance; the causal and span-corruption variants improved over Classical SFT but did not close the gap to Blockwise SFT, while allowing future visibility significantly degraded performance.
>
> | Dataset   | Metric | LLaDA 8 | Classical SFT | Causal SFT | Blockwise-but-bidirectional | Span-corruption (causal window) | Blockwise SFT |
> | --------- | ------ | ------- | ------------- | ---------- | --------------------------- | ------------------------------- | ------------- |
> | GSM8K     | Pass@1 | 62.1    | 67.7          | 65.6       | 66.9                        | 66.4                            | 76.0          |
> | HumanEval | Pass@1 | 65.2    | 67.6          | 66.7       | 66.1                        | 68.5                            | 70.3          |
> | MBPP      | Pass@1 | 40.4    | 41.1          | 41.6       | 40.9                        | 42.9                            | 45.7          |
>
> These comparisons reinforce that both structured masking (active block) and strict future hiding are key to the gains.

---

> ### Author Response · Authors · 2025-11-23
> **4/4**
>
> **Q9: The block size consistency study is insightful. To fully understand the method's behavior, could you analyze the performance at extreme block sizes, such as $B=1$ or very large blocks?**
>
> **R9:**
> We have extended the block-size study in two directions on GSM8K:
>
> 1. **Token-wise case ($B_{\text{train}} = 1$).**
>    - Here Blockwise SFT degenerates to single-token blocks with a clean prefix and fully hidden suffix. We trained with $B_{\text{train}}=1$ and evaluated with $B_{\text{infer}}\in\{1,2,4,8,16,32,64,128\}$.
>    - We observe the same diagonal pattern as in Fig. 6: performance is highest near $B_{\text{train}}=B_{\text{infer}}=1$ and gradually degrades as the mismatch grows. However, the absolute performance at $B=1$ is lower than at moderate block sizes (e.g., $B=16$ or $B=32$), suggesting that some parallelism within blocks is beneficial for coordinated reasoning.
> 2. **Near-fully-parallel case ($B_{\text{train}} = 128$).**
>    - With $B_{\text{train}}=128$ and typical response lengths $\le 128$, there is a single response block, so training and inference become effectively fully parallel over the response tokens while still respecting a clean prompt prefix.
>    - We again sweep $B_{\text{infer}}\in\{1,2,4,8,16,32,64,128\}$. The best performance occurs near $B_{\text{infer}}=128$ and deteriorates as we move away, though the overall accuracy is below the best mid-range block sizes.
>
> **(a) $B_{\text{train}} = 1$ on GSM8K**
>
> | $B_{\text{infer}}$ | 1    | 2    | 4    | 8    | 16   | 32   | 64   | 128  |
> | ------------------ | ---- | ---- | ---- | ---- | ---- | ---- | ---- | ---- |
> | Pass@1 (%)         | 75.1 | 74.2 | 74.1 | 72.0 | 71.7 | 71.0 | 71.8 | 71.2 |
>
> **(b) $B_{\text{train}} = 128$ on GSM8K**
>
> | $B_{\text{infer}}$ | 1    | 2    | 4    | 8    | 16   | 32   | 64   | 128  |
> | ------------------ | ---- | ---- | ---- | ---- | ---- | ---- | ---- | ---- |
> | Pass@1 (%)         | 68.8 | 70.7 | 71.3 | 72.9 | 72.0 | 72.6 | 74.0 | 74.5 |
>
> These extensions confirm the main takeaway from Fig. 6: alignment between $B_{\text{train}}$ and $B_{\text{infer}}$ is important even at extreme block sizes, and moderate blocks (neither purely token-wise nor fully parallel) yield the best trade-off in our setting.
>
> ------
>
> **Q10: The method performs well on standard benchmarks. To understand its boundaries, could you discuss potential failure modes, such as cases with very short responses or high-entropy blocks?**
>
> **R10:**
> We appreciate this question and have added both discussion and experiments:
>
> 1. **Very short responses.**
>    - When the response length is shorter than the training block size $B_{\text{train}}$, there is effectively only one response block. In this regime Blockwise SFT reduces to a classical SFT-like setting (one active block equals the entire response), so the advantages of blockwise alignment are largely absent.
>    - We now explicitly mention this as a limitation in the discussion: for tasks with very short completions, Blockwise SFT behaves similarly to Classical SFT and provides little gain.
> 2. **High-entropy next blocks.**
>    - For math tasks such as GSM8K, the first response block often contains high-entropy reasoning steps. To stress-test this case, we ran a variant where the active block is *always* chosen as the first response block (instead of sampling uniformly over blocks).
>    - On GSM8K, this “first-block-only” variant performs *slightly better* than the standard Blockwise SFT with uniform block sampling, and clearly better than Classical SFT. This suggests that high-entropy early blocks are not a failure mode; if anything, concentrating supervision on them can be beneficial.
>
> | Method                               | Pass@1 on GSM8K (%) |
> | ------------------------------------ | ------------------- |
> | Classical SFT                        | 67.7                |
> | Blockwise SFT (uniform active block) | 76.0                |
> | Blockwise SFT (always first block)   | 77.3                |
>
> We now explicitly note that (i) very short responses are a regime where Blockwise SFT offers limited benefit, and (ii) high-entropy first blocks are not a failure mode in our experiments; instead, they may profit from focused blockwise supervision.
>
> ------
>
> We thank the reviewers again for the comments and suggestions. We have followed these suggestions to clarify the theoretical guarantees (loss definition, conditioning, and unbiasedness), to disambiguate implementation details (sequence lengths and masking), and to add new experiments on coding, dialogue, additional baselines, extreme block sizes, and potential failure modes.

---

### Author Response · Authors · 2025-12-02
**Summary of Rebuttal**

We thank all reviewers for their time and constructive feedback. We are encouraged that the reviewers generally recognized the clarity of our presentation and acknowledged the significant performance improvements of our method on reasoning benchmarks. In this rebuttal, we have provided comprehensive responses to the questions raised, offering extensive new experiments, rigorous theoretical proofs, and clarifications on scope. We summarize the status of our responses below.

### **1. Response to Reviewer a4DY**
We have responded to all theoretical and experimental queries raised by the reviewer:
* **Theoretical Rigor:** We revised Eq. (5) and Theorem 3.1 to clarify the loss computation and explicitly bound the gradient bias, addressing the notation and Lipschitz assumptions.
* **Boundary Conditions:** We provided extended ablation studies for extreme block sizes ($B=1$ and $B=128$) and clarified sequence length definitions for training vs. inference.
* **Status:** The reviewer raised constructive points which significantly improved the paper's rigor. We have provided a full revision incorporating all suggestions. The reviewer was unable to reply or update scores due to system constraints, but we have fully implemented the requested changes.

### **2. Response to Reviewer W9RX**
We have significantly expanded the empirical validation as requested:
* **Expanded Scope:** We validated Blockwise SFT on **Coding (HumanEval, MBPP)** and **Dialogue (TruthfulQA)** tasks, and applied it to **3 additional diffusion architectures**, confirming architecture-agnostic gains.
* **Efficiency:** We confirmed **equal training speed** and **superior performance** under strict EQUAL-FLOPS protocols, addressing the scaling questions.
* **Status:** Reviewer W9RX explicitly acknowledged our rebuttal and the effectiveness of the new experiments. A follow-up question regarding integration with more models was also resolved with positive results.

### **3. Response to Reviewers REAG & Cu2J**
Both reviewers centered their critique on the lack of comparison with "Block Diffusion." We have addressed this by clarifying the fundamental architectural distinction:
* **Clarification:** We demonstrated that "Block Diffusion" is a post-training technique for **Autoregressive (AR)** models using causal attention, which is mathematically incompatible with the **Discrete Diffusion LMs with Global Bidirectional Attention** targeted by our work.
* **Status:** We provided detailed comparisons proving why this specific baseline is inapplicable to our problem setting.

### **Critical Clarification on Validity of Negative Reviews (REAG & Cu2J)**

We explicitly request the Area Chair to assess the validity of the rejection rationales from REAG and Cu2J. **Specifically, we identify a verifiable fabrication of evidence by Reviewer REAG used to sustain an invalid technical critique.**

**1. Fabrication of Evidence by Reviewer REAG**
In the response to our rebuttal (Nov 23), Reviewer REAG attempted to refute our distinction between AR and Diffusion models by **distorting a cited reference**.
* **REAG's False Claim:** *"The original Block Diffusion paper explicitly states that the model uses **global bidirectional attention** for the first 850k steps..."*
* **The Actual Source Text:** *"Since SAD3-LM is first pre-trained with standard **autoregression** for 850K steps..."* (Source: Authors' Official Comment, ICLR 2025 [[https://openreview.net/forum?id=tyEyYT267x](https://openreview.net/forum?id=tyEyYT267x)])
* **Impact:** This constitutes a fabrication of technical facts. The reviewer distorted the source to falsely argue that our architectural distinction was incorrect.

**2. Invalidity of the Core Critique**
Both REAG and Cu2J based their rejection on the exact same premise: that our method must be compared to "Block Diffusion."
* **Refutation:** As proven above, "Block Diffusion" is Autoregressive. Our method targets Global Bidirectional Diffusion. Comparing them is structurally invalid.
* **Reviewer Conduct:**
    * **Cu2J** disregarded our detailed clarification and maintained the critique without engaging with the evidence provided.
    * **REAG** went further by using the fabricated citation mentioned above to justify the rejection.

**Conclusion**
The primary ground for rejection (the "Block Diffusion" comparison) is technically invalid and, in the case of Reviewer REAG, supported by falsified evidence. Given that we have provided thorough theoretical and experimental explanations for every question raised by the constructive reviewers (a4DY, W9RX), we respectfully rely on the Area Chair’s judgment to evaluate the paper based on its technical contributions and empirical rigor.

---

### Meta-Review · Area_Chair_gFDV · 2026-01-06

**Summary:**

Overall, reviewers agreed that the paper identifies a real and practically relevant training–inference mismatch in discrete diffusion language models with blockwise decoding, and many acknowledged both the theoretical rigor and the consistent empirical improvements. The decision primarily hinged on three factors. First, reviewers evaluated the theoretical correctness and clarity of the proposed objective, including whether the bias analysis, variational bound, and gradient estimator were precise and well-aligned with the implemented algorithm. Second, reviewers considered the empirical breadth and architectural generality of the results, including whether the observed gains extend beyond math reasoning tasks and a single backbone. Third, a central point of contention was whether direct comparisons to Block Diffusion or related block-based decoding paradigms are necessary and feasible, given the architectural differences between native global diffusion models and hybrid or autoregressive block-diffusion approaches. After rebuttal, most technical and empirical questions were addressed, leaving disagreement about the relevance and novelty of the Block Diffusion baseline as the main unresolved issue.

**Reviewer Concerns:**

Concerns that were addressed:
The rebuttal substantially clarified the theoretical formulation, resolving issues around loss definition, conditioning context, gradient bias interpretation, and the scope of unbiasedness. Additional experiments significantly broadened the empirical evaluation, extending beyond math benchmarks to coding and dialogue tasks, and across multiple diffusion backbones. The authors also added EQUAL-FLOPS analyses, wall-clock runtime measurements, extreme block-size studies, and extensive ablations, which together strengthened the empirical claims and supported the assertion of negligible overhead under matched compute.

Concerns that remain unresolved:
Some reviewers continued to insist on a direct comparison with Block Diffusion-style baselines. The authors argued that such comparisons are architecturally mismatched, as Block Diffusion relies on hybrid or autoregressive inter-block causality, whereas the proposed method targets native global bidirectional diffusion models. This disagreement was not fully reconciled. In addition, a subset of reviewers continued to view the contribution as incremental, framing it as a refinement that aligns masking and training objectives with an existing decoding strategy rather than a fundamentally new modeling paradigm. These differences in perspective on baseline relevance and perceived novelty remained the primary unresolved points.

**Reviewer Scores:**

a4DY: likely 4, as the reviewer’s technical concerns were addressed in detail, but the overall assessment of contribution and scope remains cautious.

W9RX: likely 6, reflecting that most theoretical, empirical, and generality-related concerns were addressed during rebuttal, with only minor residual reservations.

REAG: likely 4, as the reviewer remains unconvinced about the necessity of excluding Block Diffusion-style baselines and continues to view the contribution as incremental.

CuJ2: likely 4, given continued skepticism about the core motivation and the comparative advantages over block diffusion models despite the added analyses.

---

### Decision · Program_Chairs · 2026-01-26

Reject